# SUMMING UP THE FACTS: ADDITIVE MECHANISMS BEHIND FACTUAL RECALL IN LLMS

## ABSTRACT

How do large language models (LLMs) store and retrieve knowledge? We focus on the most basic form of this task – factual recall, where the model is tasked with explicitly surfacing stored facts in prompts of form `Fact: The Colosseum is in the country of`. We find that the mechanistic story behind factual recall is more complex than previously thought – we show there exist several distinct, independent and qualitatively different mechanisms that additively combine, constructively interfering on the correct attribute. We term this generic phenomena the **additive motif**: models compute correct answers through adding together multiple independent contributions; the contributions from each mechanism may be insufficient alone, but together they constructively interfere on the correct attribute when summed. In addition, we extend the method of direct logit attribution to attribute a head's output to individual source tokens. We use this technique to unpack what we call 'mixed heads' – which are themselves a pair of two separate additive updates from different source tokens.

## 1 INTRODUCTION

How do large language models (LLMs) store and use factual knowledge? We study the factual recall set up, where models are explicitly tasked with surfacing knowledge as output tokens in prompts of form `Fact: The Colosseum is in the country of`. Our work falls within the field of mechanistic interpretability (Elhage et al., 2021; Olah et al., 2020), which focuses on reverse-engineering the algorithms that trained neural networks have learned. Much attention has recently been paid to interpreting decoder-only transformer-based large language models, as while these models have demonstrated impressive capabilities (Brown et al., 2020; Wei et al., 2022), we have little understanding into *how* these models produce their outputs.

Prior work on interpreting factual recall has mostly focused on localizing knowledge within transformer parameters. Meng et al. (2023a) find an important role of early MLP layers is to *enrich* the internal representations of subjects (`The Colosseum`), through simultaneously looking up all known facts, and storing them in activations on the final subject token. Since the model is autoregressive, this occurs before seeing which relation (`country of`) is requested. We study how this information is subsequently moved and used by the model. There are several possible mechanisms models *could* use to retrieve facts from these enriched subject representations. Geva et al. (2023a) suggest an algorithm that allows models to extract just the correct fact, ignoring other irrelevant facts in the subject representation. Hernandez et al. (2023) more recently showed that such facts can be *linearly* decoded from the enriched subject representations. In this paper, we build on this prior work by carefully inspecting internal model mechanisms using tools from mechanistic interpretability.

Our core contribution in this work is showing that models primarily solve factual recall tasks **additively**. We say models produce outputs additively if

1. There are multiple model components whose outputs independently directly contribute positively to the correct logit.
2. These components are qualitatively different – their distribution over output logits are meaningfully different.
3. These components constructively interfere on the correct answer, even if the correct answer is not the argmax output logit of individual components in isolation.

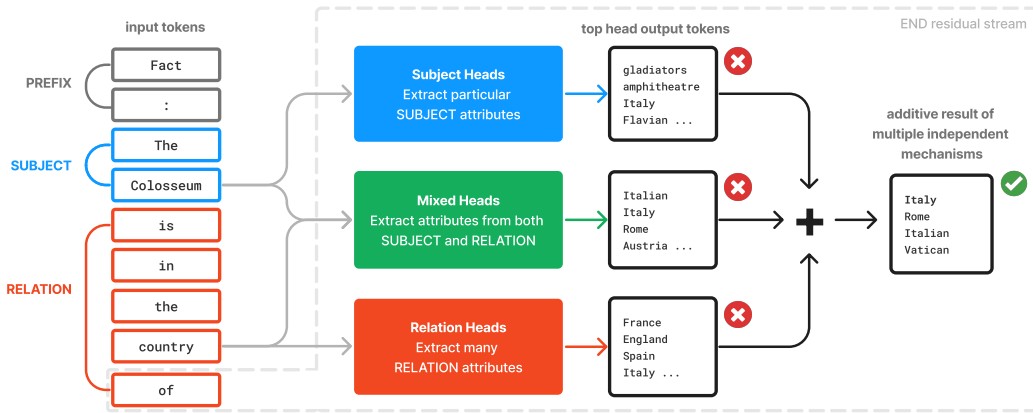

Figure 1: Four independent mechanisms models use for factual recall. (1) Subject heads, (2) Relation Heads, (3) Mixed Heads and (4) MLPs (omitted). These combine **additively**, **constructively interfering** to elicit the correct answer. Each mechanism individually is less performant than the sum of them all, with most individual mechanisms incapable of performing the task alone.

We term this generic phenomena the **additive motif**. We provide further discussion regarding this in Section 4.

Consider the example shown in Figure 1. There are two sources of information here – the subject `Colosseum` and the relation `country`. These correspond to two clusters of additive updates - updates that write many attributes about the Colosseum (e.g. `Italy`, `Rome`), and updates that write many countries (e.g. `Italy`, `Spain`). By using mechanistic interpretability to investigate the how factual recall is performed by the model, we find four different mechanisms implement these two updates. Each mechanism independently boosts the correct answer (condition 1). There are two qualitatively different clusters of output behavior (condition 2). And while each mechanism may not individually completely solve the task, we find that additively combining all four results in a large amount of constructive interference on correct attributes – this is significantly more robust (condition 3).

Our work also highlights a limitation of narrow circuit analysis. We should expect models to make predictions based on multiple parts of their input. Prior mechanistic interpretability work has neglected to consider all sources of information in mechanistic analysis. For instance, in the work by Wang et al. (2022) models are tasked with completing sentences of the form `When John and Mary went to the store, John bought flowers for`. This task has two components – (a) figure out the answer should be a name, and then (b) figure out what the correct name is. By using the 'logit difference' `Mary - John` as a metric, the authors isolate the circuit for (b), but neglect to study (a). Though just studying (b) and conditioning on the answer being a name is a valid research question, it's important to be explicit that part of the behaviour is left unexplained, and our work implies that (a) is also an important part of predicting the next token. In factual recall, this corresponds to updating outputs based on the relation, as well as the subject. We find additivity through studying both of these sources of information, and analyzing output attributes relating to both.

We additionally extend the technique of direct logit attribution (DLA) (Wang et al., 2022; Elhage et al., 2021; nostalgebraist, 2020). DLA is a technique that converts individual model component (attention head, MLP neuron) outputs into the space of output logits, through the insights that the map to logits from the residual stream is linear,[1] and that the residual stream is a cumulative sum of prior model components (Elhage et al., 2021). DLA by default considers the entire attention head as one unit, but Elhage et al. (2021) demonstrate that attention head outputs are a linear weighted sum over source positions. We may therefore split the DLA of attention head up into contributions from different source tokens. This insight allows us to disentangle the two separate and additive contributions of particular attention heads from SUBJECT and RELATION tokens.

---

[1]up to LayerNorm, which may be reduced to just a scaling factor for our purposes (Nanda, 2022).

| Subject $s$ | Relation $r$ | Attribute $a$ | Attributes $S\backslash\{a\}$ | Attributes $R\backslash\{a\}$ |
|---|---|---|---|---|
| Kobe Bryant | plays the sport of | basketball | NBA, Lakers, USA | tennis, golf, football |
| The Eiffel Tower | is in the country of | France | Paris, iron, Gustave | Pakistan, China, Sudan |
| Germany | has capital city | Berlin | German, Rhine, BMW | London, Rome, Canberra |

Table 1: Some examples of factual tuples $(s, r, a)$. We prepend the prefix `Fact:` to the concatenated pair $(s, r)$ for inference, as this slightly improves performance. We also include example elements in the sets $S$ and $R$ of attributes pertaining to the subject $s$ and relation $r$ respectively.

## 2 METHODOLOGY

**Task**. We consider tuples $(s, r, a)$ of factual information containing a subject $s$, attribute [2] $a$, and relation $r$ connecting the two. To elicit facts in models, we provide a natural language prompt describing the pair $(s, r)$. See Table 1 for example tuples and prompts. At various points we study and aggregate over sets $(s, r, a)$ with $s$ or $r$ held constant. We filter for tuples $(s, r, a)$ for which the model attains the correct answer, which we define as $a$ being within the top ten output tokens. Most commonly the correct attribute $a$ attains rank 0 (See Figure 7 in the Appendix). Our dataset is hand written, but is inspired by `CounterFact` (Meng et al., 2023a), `ParaRel`(Elazar et al., 2021), and Hernandez et al. (2023). See Appendix C for more information on our dataset, including a discussion of issues we encountered in creating this dataset.

**Model**. We primarily investigate the Pythia-2.8b model Biderman et al. (2023), though find similar mechanisms are present in other models (GPT2-XL (Radford et al.), GPT-J (Wang and Komatsuzaki, 2021), and other Pythia models), which we discuss briefly in Appendix E.2.

**Counterfactual Attributes**. We are interested in what mechanisms surface the correct attributes a. In order to better understand this, we find it useful to study two further sets of attributes[3] $S$ and $R$. The correct attribute $a \in S \cap R$. $S$ is the set of attributes relevant to the subject. In particular, an attribute $a \in S$ if there exists some other relationship $r'$ such that $(s, r', a)$ is a valid factual tuple. $R$ is the set of attributes relevant to the relation. An attribute $a \in R$ if there exists some other subject $s'$ such that $(s', r, a)$ is a valid factual tuple. See Table 1 for example elements in $S$ and $R$.

**Token Positions**. We will often refer to particular token positions of interest.

- PREFIX– all tokens *before* the subject, usually `Fact:`
- SUBJECT– all tokens of the subject $s$, e.g. `The Colosseum`.
- RELATION– all (other) tokens of the relation $r$ e.g. `is in the country of`.
- END– the final token, which is where factual information must be moved to in order to surface the correct answer, e.g. `of`.

**Logit Lens.** The logit lens (nostalgebraist, 2020) is an interpretability technique for interpreting intermediate activations of language models, through the insights that the residual stream is a linear sum of contributions from each layer (Elhage et al., 2021) and that the map to logits is approximately linear. It pauses model computation early, converting hidden residual stream activations to a set of logits over the vocabulary at each layer.

**Direct Logit Attribution** (DLA) is an extension of the logit lens technique. It zooms in to individual model components, through the insight that the residual stream of a transformer can be viewed as an accumulated sum of outputs from all model components (Elhage et al., 2021).[4] DLA therefore gives a measure of the direct effect on the of individual model components on model outputs. We extend this technique to **DLA by source token group** through the further insight that attention head outputs are a weighted sum of outputs corresponding to distinct attention source position (Elhage et al., 2021), allowing us to quantify how each group of source tokens in turn contributes directly to

---

[2]We use the words 'attribute' and 'fact' interchangeably.

[3]While our sets may not be complete, or faithful to true model concepts, but do suffice to help us find mechanisms.

[4]DLA can be limited, see e.g. Rager et al. (2023).

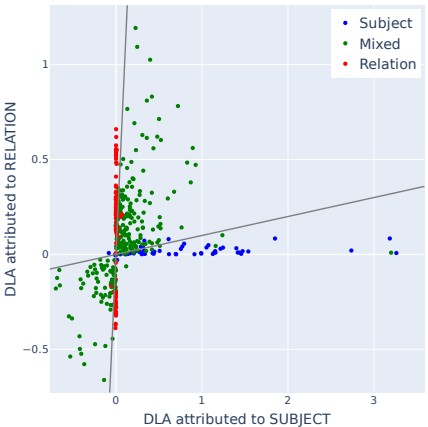 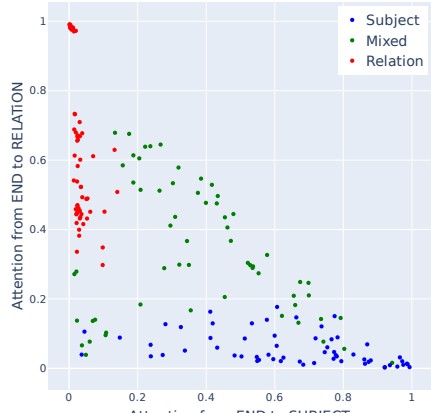

Figure 2: Three different types of attention head for factual extraction for prompts of form `s plays the sport of`: subject heads, relation heads and mixed heads. (**Left**) DLA on the correct sport, split by attention head *source* token, for top 10 heads by total DLA. Each data point is one prompt for one factual tuple. The grey lines have gradients $1/10$ and $10$ and denote the boundary we use to **define** head types, *after* aggregating over a relation $r$. These cleanly separate subject and relation heads. (**Right**) Attention patterns of the top four heads of each kind on each prompt in the dataset. Subject and Relation heads attend mostly to SUBJECT and RELATION respectively. Mixed heads attend to both. Attention patterns is not used to define head type, but correlate well.

the logits. This is useful in disentangling head types, in particular mixed heads (Figure 1), which comprise two separate contributions from their attention paid to the subject and their attention paid to the relationship. See Appendix D for more details on this technique. We say the DLA can be *attributed* to either the SUBJECT tokens or RELATION tokens. This mostly makes sense for the short prompts in our setup, but may be misleading in longer context lengths, as models move information around and may store information on intermediate tokens.

## 3 RESULTS

In this section, we use mechanistic interpretability to find four separate mechanisms behind factual recall that correspond to two clusters of additive updates, relating to either the subject or relation in the prompt. These updates constructively interfere on the correct attribute to elicit the correct answer. These mechanisms all act on the END position. We summarize these mechanisms as follows and in Figure 1.

1. **Subject Heads** (Section 3.1) – Attention Heads that attend strongly to SUBJECT and extract attributes pertaining to the subject, in the set $S$, from the enriched subject representation. Some such heads extract the correct attribute $a$, others extract a range of other attributes. These heads activate in response to any factual recall type prompt, even if the relationship given does not match their category - they can and do *misfire*, extracting irrelevant attributes.
2. **Relation Heads** (Section 3.2)– Attention Heads that attend strongly to RELATION for a particular relation and extract many attributes pertaining to the relation, in the set $R$. They do not preferentially extract the correct attribute associated with the subject, $a$.
3. **Mixed Heads** (Section 3.3)– Attention Heads that attend to both SUBJECT and RELATION, and perform the role of both (1) and (2) simultaneously. From SUBJECT, they extract attributes the correct attribute $a$, among other things. From RELATION, they extract many attributes in the set $R$, often also privileging the correct attribute $a$, due to a phenomena we term 'subject to relation propagation'. The sum of these two separate contributions is the total head direct effect.
4. **MLPs** (Section 3.4)– Part of the function of MLPs is to boost many attributes in the set $R$.

Inspecting the logit ranks is highly suggestive of an additive algorithm: many incorrect attributes in both of the sets $S$ and $R$ appear highly in output tokens (Table 4 in the Appendix).

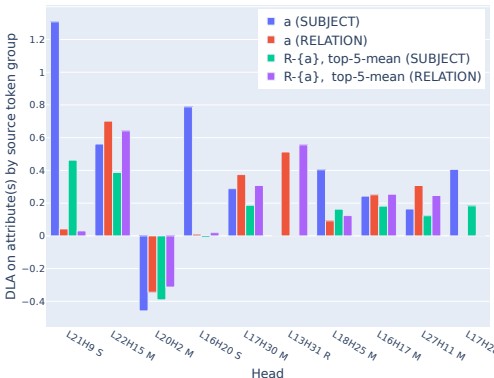 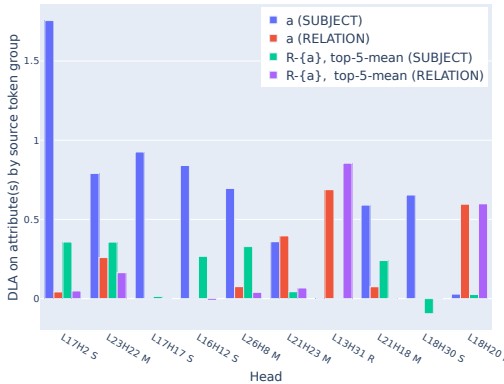

Figure 3: Top heads by absolute DLA for the relationships `plays the sport of` (**left**) and for `is in the country of` (**right**), labeled as Subject (S), Relation (R) or Mixed (M) heads. Studying a large set of counterfactual attributes, and splitting by attention source token lets us disentangle these head types. We plot DLA on the correct attribute $a$. We also plot the mean DLA on the 5 largest magnitude attributes in $R - \{a\}$. We split all DLAs by attention source token (SUBJECT vs RELATION). All three head types emerge. Subject heads are characterized by the largest column being blue. Relation heads have comparable red and purple columns, with small blue and green columns. Mixed heads capture everything remaining.

In the remainder of this section, we provide several lines of evidence that these four mechanisms implement an additive algorithm. In particular, we will show (a) all four mechanisms exist for a range of relationships and are distinct, (b) each mechanism contributes positively to both correct and incorrect attributes and matters for task performance and therefore (c) each individual mechanism is inferior to the sum of all four mechanisms. Showing (a-c) suffices via our definition of additivity in the Introduction. We perform further experiments in Appendix E. Figures 2 and 3 summarise these results.

## 3.1 SUBJECT HEADS

Individual subject heads extract specific attributes about subjects in some set $S \cap C$ by attending from END to SUBJECT, but not meaningfully to RELATION. [5] These heads extract the same attributes from a given subject *no matter what relationship is given* – the attribute `basketball` is still extracted significantly by some subject head on the prompt `Michael Jordan is from the country of`. Such heads explain why we observe incorrect attributes about the subject (i.e. in the set $S$) appearing in the top few output tokens on factual recall prompts. These heads sometimes depend on the relationship indirectly, through their attention pattern.

We define subject heads for a relation $r$ to be heads with average DLA attributed to SUBJECT tokens / average DLA attributed to RELATION tokens > 10, when aggregated over a dataset of prompts with the relation held constant. This captures the intuition that these heads primarily read attributes from the subject and not the relation.

In Figure 4, we analyze subject heads for different relationships across a range of subjects. We may view individual heads as linear probes for particular output tokens through their $OV$ (output-value) circuit (Elhage et al., 2021), which projects the final SUBJECT token's residual stream to unembedding directions, which we map to logits. This technique effectively saturates the attention of the subject head to one on the final subject token, performs the usual attention head calculation, and reads off some DLA from the output. Since subject heads *always* attend to the subject, this is principled: we discuss attention patterns of subject heads in Appendix E.4. We evaluate each head-probe qualitatively on a range of subjects, showing they extract meaningful and interpretable attributes from the enriched subject representation. This also demonstrates that the head category $C$

---

[5]Generically, since individual attention heads read and write from a low rank subspace of the residual stream (Elhage et al., 2021), we find them to be specialized to same category of attributes $C$, which may not perfectly align with $S$ or $R$ (See Appendix E.8 for more discussion on head categories).

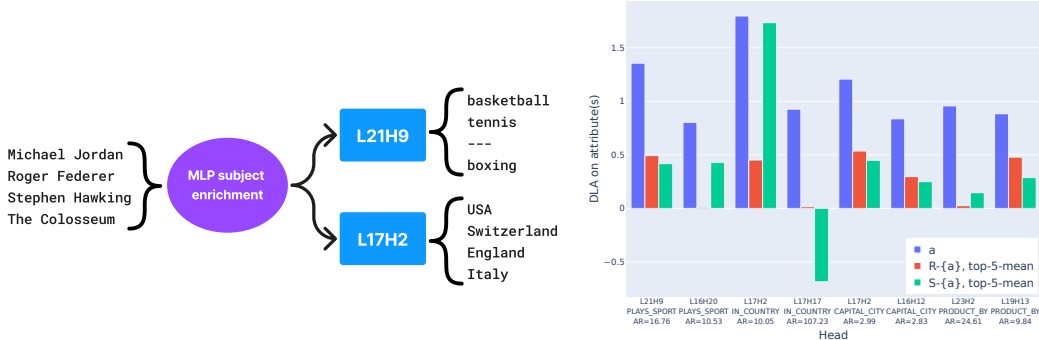

Figure 4: Subject Heads exist for a range of relations. (**Left**) The mechanism by which subject heads act. They read from enriched subject representations, and copy the relevant attributes to output directions. We show this for a 'sport' head and a 'country' head. Both pathways activate whenever a factual recall prompt with the given subject is presented, no matter what the stated relationship is. No sport is extracted for `Stephen Hawking`. Raw data for this figure is in the Appendix in Table 6. (**Right**) Top two subject heads for four different relationships. These heads individually extract the correct attribute (blue) significantly more than other relation attributes $R$ (red) and other subject attributes $S$ (green). This indicates their category $C$ is mostly narrow. L17H2 is more general, extracting many correlated facts about countries (e.g. country, currency, cities, etc.). These heads also have a high attention ratio to SUBJECT over RELATION (shown in the x axis labels).

is not aligned with $R$ or $S$: e.g. L22H17 extracts only the sport of `basketball`, but not other sports. We also observe correlated facts `NBA` and `basketball` being extracted simultaneously.

## 3.2 RELATION HEADS

Individual relation heads extract many attributes in the set $R \cap C$ by attending from END to RELATION, but not significantly to SUBJECT tokens. These heads do not causally depend on the subject, even indirectly. Such heads explain why we observe incorrect attributes pertaining to the relation (i.e. in the set $R$) appearing in the top few output tokens on factual recall prompts.

We define relation heads for a relation $r$ to be heads with average DLA attributed to RELATION tokens / average DLA attributed to SUBJECT tokens > 10, over a dataset of prompts with the the relation held constant. This captures the intuition that relation heads mostly read the correct attribute from the relation, and not the subject.

Figure 5 demonstrates relation heads exist for a range of relationships $r$ and that their direct effect on logits mostly does not depend on the subject $s$. Preliminary results suggest this latter finding is less true in larger models; a result which we expand on in Appendix E.5.4. These heads can additionally be characterized through high attention to the RELATION over SUBJECT. Interestingly, there are many shared heads between relationships, including L13H31, which is important for both sports and countries. Each relation head will push for certain attributes over others, with a small amount of variation from prompt to prompt. Which attributes a relation head prefers is affected minimally by the subject. A complication is that DLA can be affected by the norm of the accumulated residual stream (via LayerNorm), which varies slightly between prompts, leading to some variation.

To show this is a large effect, we analyze the ordered DLA across all vocab tokens of the top few relation heads for several prompts in Appendix E.5. This demonstrates that the *primary function* of these heads is to extract attributes in $R$. We also perform patching experiments, where we patch the top few relation heads, and demonstrate that this does not reduce performance on average - indicating that these heads do not meaningfully depend on the subject, even indirectly.

## 3.3 MIXED HEADS

Individual mixed heads extract many attributes in some set $(S \cup R) \cap C$, and also privilege the correct attribute $a$ among such attributes. They behave as a combination of subject and relation heads – they attend to both SUBJECT and RELATION. From SUBJECT, they extract the correct attribute

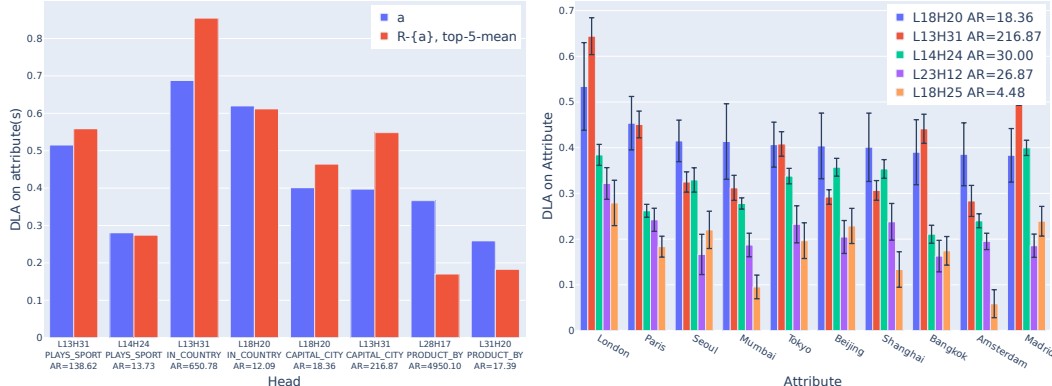

Figure 5: Relation heads exist for a range of different relationships. (**Left**) The top two relation heads for four different relationships. The heads extract the correct attribute (blue) about as much as they extract many other attributes in the set $R$ (red). They also have a high attention ratio to RELATION over SUBJECT (shown in the x axis labels). (**Right**) Many cities are extracted by heads over a range of prompts with relation `has the capital city` with *different* subjects. The error bars denote the standard deviation over these subjects. While heads push for some cities more than others, small error bars indicate this variation is consistent across input subjects. This suggests relation head outputs do not causally depend on the subject. We include similar plots for other relationships in Appendix E.5.

$a$ more than other attributes from $R$. From RELATION, they extract many attributes in $R$, often also privileging $a$. This is due to significant propagation of subject information to the RELATION, which we do not rigorously study, but attempt to disentangle in Appendix E.6. We attribute the two contributions from different source positions SUBJECT and RELATION through our DLA by source technique.

Figure 3 demonstrates this; We see mixed heads generally extract the correct attribute $a$ from *both* SUBJECT and RELATION (blue and green) more than other relation attributes $R$ (red and purple). To further illustrate this effect, we analyze the top DLA token outputs of a selection of mixed heads in the Appendix in Table 8, split by source token, demonstrating these heads (a) attend to two distinct places and (b) extract significant information from these two distinct places.

### 3.4 MLPs

MLP layers on the END token often uniformly boost many attributes in the set $R$ (like relation heads). The MLPs do not preferentially boost the correct attribute $a$. We find that the category $C$ of the MLP direct effect is significantly larger than those of individual heads, which intuitively makes sense given the MLP has many more parameters than individual attention heads. Individual neurons would likely have much more restricted categories. We note we only study part of the function of the MLP, and only on the END position. We hypothesize MLPs either compose with relation heads, or with relation information directly.

In Figure 6, we show that for a range of relationships an aspect of the total direct effect of the MLP layers is to boost many attributes in $R$, including $a$, but that $a$ is not privileged among the attributes $R$. We too see that while the MLP layers up-weight certain attributes more than others, this variation is consistent across subjects, indicating these outputs do not causally depend on the subject. In Appendix E.7, we show that, at least for some relationships, this is the primary direct effect of the MLP layers, through analyzing the top DLA tokens of summed MLP outputs. See Appendix E.7 for more experiments.

## 4 Discussion

**Additivity.** We speculate that models in general prefer to solve tasks in an *additive* manner via multiple independent circuits, as we describe in Section 1. This claim is supported by prior work (Nanda et al., 2023; Chughtai et al., 2023). We do not explain *why* the additive mechanism is preferred,

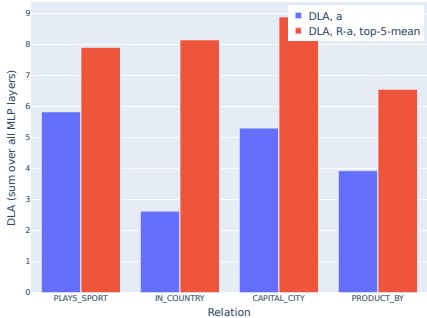 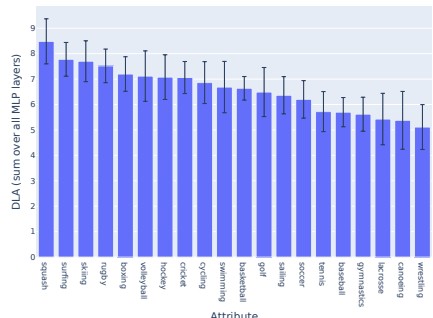

Figure 6: (**Left**) The sum of all MLP outputs boosts relation attributes $R$ for a range of relationships. The MLPs boost the correct attribute (blue) less than they boost other attributes in the set $R$ (red). The MLP boosts a wider set of attributes in $R$ than we automatically check: this is a limitation of our dataset, and would not be the case with a wider dataset (see Table 10 in the Appendix). (**Right**) many sports are boosted by MLPs over a range of prompts with relation `plays the sport of`, independent of which subject is given. Error bars are standard deviation over different subjects. This suggests the direct effect of the MLP does not causally depend on the subject.

but speculate that compounding evidence through several simple circuits is significantly easier for models than more complex alternatives. Due to a softmax being applied to model outputs when taking cross-entropy loss, models extremize their outputs. Outputting small amounts of incorrect answers is therefore not that costly to the model, so long as constructive interference results in a large logit difference between correct and incorrect answers pre-softmax.

As additional intuition, we present a toy example of additivity. Consider a logistic model tasked with predicting whether an integer is divisible by 6 (into two classes, true or false). Consider the following two mechanistic ways of solving the task. (a) Solve the task directly, memorising which integers are divisible by 6. (b) Solve the task in two independent parts. Assign +1 true logit to all numbers divisible by 2. Assign +1 true logit to all numbers divisible by 3, with a different circuit. Apply a uniform bias corresponding to a -1.5 false logit. Both mechanisms solve the task.

In this example (a) is non additive. (b) is additive, by the criteria (1-3) given in Section 1. There are two different components that contribute to the answer (1), they have qualitatively different outputs (2), which constructively interfere on the correct answer, with each component insufficient alone (3). This example is analogous to how a transformer functions, since the residual stream is an additive sum of outputs from model components, and there iss an (approximately) linear map from the residual stream to the output logits given by the unembedding, so each component can be considered to be writing to logits separately in a linear fashion (Elhage et al., 2021). Note that condition (2) is necessary to exclude cases where the model increases its confidence through adding two identical components, which we do not consider to be additive.

**Reversal Curse**. Our work on factual recall offers a mechanistic explanation for the reversal curse – the noted limitation of LLMs to generalize to 'B is A' when trained on 'A is B' (Berglund et al., 2023), which has also been suggested by Grosse et al. (2023); Thibodeau (2022). In our work, we provide indirect and suggestive evidence this is to be expected. We find a circuit by which models may learn to output 'A is B', involving subject enrichment on the A tokens, and some attention head attending to A and extracting B. Importantly, this is a unidirectional circuit with two unidirectional components - it extracts the fact 'B' from 'A'. Our circuit suggests that the reason fine-tuning on A is B does not boost 'B is A' in general is because training on 'A is B' only boosts the unidirectional A $\rightarrow$ B mechanisms, and has no effect on potential B $\rightarrow$ A mechanisms. As further evidence, assembling multi-token input representations is a different task mechanistically to outputting multi-token facts. This is in part due to input and output spaces being separate – Embeddings and unembeddings are untied in modern LLMs: $W_E \neq W_U^T$. So the 'A' in 'A is B' is internally represented *differently* to the 'A' in 'B is A', further suggesting these two tasks are separate. We view this as evidence that our work, and mechanistic interpretability more generally, can produce useful insights into the kinds of high level behavior neural networks may implement.

## 5 RELATED WORK

**Interpreting Factual Knowledge.** There has been much interest in understanding and editing factual knowledge in language models in a white box manner. Geva et al. (2021) demonstrated transformer MLP layers can be interpreted as key-value memories, and later extended this to show a partial function of transformer MLP layers is to perform computation to iteratively update the distribution over output vocabulary space (Geva et al., 2022).

In a separate line of work, Meng et al. (2023a) found a separate function of MLP layers: to enrich the representations in the residual stream of subjects with facts for the model to later use, which was discovered using a causal intervention based methodology. They also had success with using this localization to edit the weights of the model to change output predictions, which was later scaled up to 10000 facts (Meng et al., 2023b). Subsequent work has demonstrated this technique may just be introducing a "loud" fact (Thibodeau, 2022), and that the performance of editing in a layer may not be a reliable way to localize the fact (Hase et al., 2023).

Equipped with this knowledge, an interesting question is that of how specific knowledge about a subject is isolated from other knowledge. Geva et al. (2023b) describe a circuit for factual recall with three steps: (1) subject enrichment in MLP sublayers, as in ROME, (2) relation propagation to the END token, and (3) selective attribute extraction by later layer attention heads. Our work offers a fuller understanding of this circuitry by zooming in more deeply into what individual model components are doing. This goes beyond the ablation based approach taken in Geva et al. (2023a). Separately, Hernandez et al. (2023) demonstrate that facts can be linearly decoded from the enriched subject residual stream, which supports an aspect of the full picture we find. We build on this by zooming in to the actual transformer mechanisms, finding linear decoding maps in head $OV$ circuits.

**Extracting Knowledge from LMs**. The standard approach to understand what a model knows is through prompting models in a black box fashion. (Petroni et al., 2019; Jiang et al., 2020; Roberts et al., 2020; Zhong et al., 2021). Elazar et al. (2021) study whether factual knowledge generalizes across paraphrasing. Our work gives initial insights into what mechanisms could explain when models may generalize to paraphrases and when they would not. Recently, Berglund et al. (2023) discuss a phenomena named the 'reversal curse', where models trained on "A is B" fail to generalize to "B is A", which has also been observed by prior work (Grosse et al., 2023; Thibodeau, 2022). Our work explains why this phenomenon is to be expected mechanistically – facts are stored as asymmetric look up tables in models, and so training on "A is B" is unlikely to reinforce the inverse look up table "B is A" too.

**Mechanistic interpretability** encompasses understanding features learned by machine learning models (Olah et al., 2017), mathematical frameworks for understanding machine learning architectures (Elhage et al., 2021), and efforts to find circuits in models (Cammarata et al., 2021; Nanda et al., 2023; Chughtai et al., 2023; Heimersheim and Janiak; Wang et al., 2022). Mechanistic interpretability work encompasses manually inspecting model components, performing causal interventions to localize model behavior (Chan et al.; Geiger et al., 2022; 2021) and work on automating the discovery of causal mechanisms (Conmy et al., 2023; Bills et al., 2023). We make use of mechanistic interpretability techniques and frameworks in this paper.

## 6 CONCLUSION

In this work, we analyze neural circuitry responsible for the recall of known facts about subjects. We show that factual recall mechanistically comprises several distinct moving parts. We find several simple and distinct mechanisms that interact **additively** to extract facts. These constructively interfere to produce the correct answer. Each mechanism is insufficient alone, but the summing up of several contributions is significantly more robust. We call this the **additive motif**. This motif seems core to the model's functioning in this fairly general set up, and so likely generalizes to other tasks - we see this as a promising direction of future investigation. Our work contributes to the growing literature on factual recall, and opens up several interesting new directions, discussed in Appendix B. We also highlight some of the limitations of narrow circuit analysis. By expanding our scope of study were able to uncover mechanisms for factual recall prior work had missed. We consider such study important for *comprehensively* understanding neural networks, a stated goal of the field of mechanistic interpretability (Elhage et al., 2021).

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

## A    LIMITATIONS

Our investigation attempts to present evidence that a range of mechanisms for factual recall exist within models, but does not claim to explain all such mechanisms. We present some evidence that all of these are important, but do not attempt to quantify how important each mechanism is. Further, while our separation of heads into Subject, Relation and Mixed heads is useful in understanding head function, the true picture is less clean, and where we draw the boundaries is somewhat arbitrary. In this paper, we argue that the distinction of at least Subject and Relation heads makes sense, but we acknowledge the mixed head boundary is somewhat fuzzy.

Since our focus is demonstrating a range of mechanisms exist, we primarily investigate one model, Pythia-2.8b, on a fairly small dataset. We discuss some of the limitations we faced during dataset curation in Appendix C.

Finally, in the plots in the main body of the paper, we generally focus on attributes that have high unigram frequency - (sports, countries, etc.). This makes analysis of individual model components simpler, as polysemantic heads will generally write common attributes with higher norm than less common ones. Our additive and constructive interference picture does still hold up for less common categories of word - however, often with lower logit lens significance on individual model components.

## B    FUTURE WORK

**Understanding Correlation**. Correlated features have been shown to be organize themselves into geometric patterns in toy set ups where there are more features than parameter count. This can be thought of as a form of lossy compression, and is known as superposition (Elhage et al., 2022). In our work, we found similar attention heads responsible for reading and writing very correlated features, eg. 'France' and 'Paris' or 'basketball' and 'NBA', suggesting these features are stored together in superposition. We know superposition exists in real language models (Gurnee et al., 2023), but an open problem is understanding how models perform *computation* of compressed features in superposition, overcoming issues of interference. In particular, it is unlikely that a linear method such as (Hernandez et al., 2023) could disentangle these. It is possible that constructive interference of our four mechanisms suffice to, but something more complex may be at play.

**Understanding MLP neurons**. In this work, we analyze MLPs very briefly, showing they generally boost many attributes in the set $R$. Understanding how this is done more precisely would be of interest. One could first zoom in to individual neurons, instead of MLP layers as a whole, and attempt to identify which inputs are responsible for boosting the unembedding directions of attributes in $R$. Is the relation information being used explicitly? Or do these neurons just compose directly with relation head outputs? MLP neurons remain a challenge in interpreting the algorithms implemented by transformers.

**ROME**. The ROME technique Meng et al. (2023a) is able to edit model outputs in a way that generalises across a range of prompts, but has some limitations. The localisation needed may not be precise (Hase et al., 2023), and the phenomenon of "loud facts" suggests ROME is not as surgical as initially thought Thibodeau (2022). Future work could use our understanding of the end to end mechanisms behind factual recall to try and understand how ROME works in an end-to-end manner, and explain mechanistically why these limitations exist.

**Prompting Set Up**. One could study how different prompting set ups affect the task of factual recall. For instance, how does a few shot prompt, or prompt injection of form "Never say 'Paris'. The Eiffel Tower is in the city of" work in improving or reducing performance? One could also study paraphrasing, in a similar fashion to (Elazar et al., 2021). One could compare the internal mechanisms found in this paper to those found for different prompting set ups and analyze the difference. Olsson et al. (2022) argues induction heads are important in in-context learning, but our understanding of the general phenomenon remains poor in general. Similarly, our understanding why jailbreaks such as that presented in Zou et al. (2023) occur would be productive in mitigating the prevalence of jailbreaks.

**Multi-Step Factual Recall**. Consider prompts of form `The largest church in the world is located in the city of`. Humans would solve this task sequentially, with two inference steps. However, models may be able to solve this task in one forward pass. Additivity may be able

to explain why. Investigating the mechanisms behind model performance in this task would be an interesting area of further investigation.

## C  DATASET

Our dataset is loosely inspired by Meng et al. (2023a) and Hernandez et al. (2023), but is manually generated. We found these preexisting datasets to be unsatisfactory for our analysis, due to some additional requirements our set up necessitated. We firstly required models to both *know* facts and to *say facts* when asked in a simple prompting set up, and for the correct attribute $a$ to be completely determined in its tokenized form by the subject and relationship. For example 'The Eiffel Tower is in' permits both the answer 'Paris' and 'France'. For simplicity we avoided prompts of this form. Synonyms also gave us issues, e.g. 'football' and 'soccer', or 'unsafe' and 'dangerous'. This mostly restricted us to very categorical facts, like sports, countries, cities, colors etc. We also wanted to avoid attributes that mostly involved copying, such as 'The Syndey Opera House is in the city of Sydney', as we expect this mechanism to differ substantially from the more general mechanism, and to rely mostly on induction heads (Olsson et al., 2022). Next, we wanted to create large datasets with $r$ held constant, and separately, with $s$ held constant. Holding the relation constant and generating many facts is fairly easy. But generally models know few facts about a given subject, e.g. 'Michael Jordan' is associated very strongly with 'basketball', but other facts about him are less important and well known. Certain kinds of attributes, like 'gender' are likely properties of the tokens themselves, and not likely not reliant on the 'subject enrichment' circuitry - e.g. 'Michael' and 'male'. We try and avoid these cases. We also restrict to attributes where the first attribute token mostly uniquely identifies the token – often the attribute is just a single token. If the first token of the attribute is a single character, this can be vague, so we omitted these cases. These considerations limited the size of the dataset we studied.

Here, we provide further details regarding our dataset. Our dataset comprises 106 prompts, across 10 different relations $r$. We summarise the relations we study in Table 2, and validate our primary model of study achieves high accuracy on the dataset in Figure 7.

| Relation | Relation Text | Number of Subjects | Example Subjects |
|---|---|---|---|
| PROFESSOR_AT | is a professor at the university of | 9 | Charles Darwin, Isaac Newton, Alan Turing |
| PLAYS_SPORT | plays the sport of | 15 | Tom Brady, Patrick Mahomes, LeBron James |
| PRIMARY_MACRO | has the primary macronutrient of | 11 | Potatoes, Rice, Oil |
| PRODUCT_BY | is a product by the company of | 9 | Wii Balance Board, Windows 10, Platform Controller hub |
| IN_COUNTRY | is in the country of | 7 | The Eiffel Tower, Sydney Opera House, Machu Picchu |
| CAPITAL_CITY | has the capital city of | 10 | Brazil, Spain, Russia |
| LEAGUE_CALLED | plays in the league called the | 6 | Tom Brady, Patrick Mahomes, Mookie Betts |
| FROM_COUNTRY | is from the country of | 12 | LeBron James, David Beckham, Kobe Bryant |
| IN_CONTINENT | is in the continent of | 7 | The Eiffel Tower, Sydney Opera House, Machu Picchu |
| IN_CITY | is in the city of | 7 | The Eiffel Tower, Sydney Opera House, Machu Picchu |

Table 2: The factual tuples in our dataset, aggregated over the relation $r$.

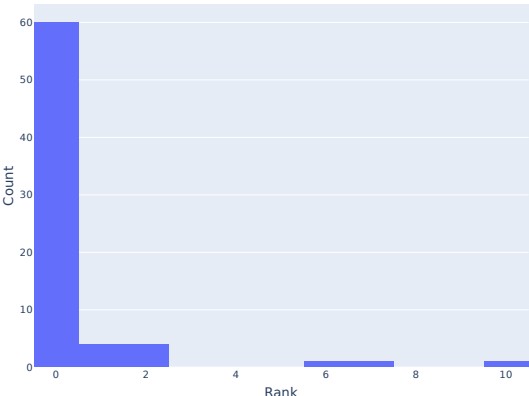

Figure 7: Ranks of the correct attribute $a$ for all prompts in our dataset on Pythia-2.9b. We filter for prompts where the attribute $a$ is within the top 10 logits. Though, the model has a very high top-1 accuracy – the rank is usually zero.

To generate sets $S$ and $R$ We used GPT-4 to generate a large list of relevant attributes for each subject $s$ and relation $r$. We then manually filtered these lists of attributes. For instance, removing attributes beginning with the token the.

## C.1 EXAMPLE DATAPOINTS

We include below three data points, corresponding to three separate tuples $(s, r, a)$, along with sets $S$ and $R$.

| subject | relation | relation text | attribute | prompt | counterfactual subject attributes | counterfactual relation attributes |
|---|---|---|---|---|---|---|
| Sydney Opera House | IN_COUNTRY | is in the country of | Australia | Fact: Sydney Opera House is in the country of | ['1973', 'Sydney', 'modern architecture', 'iconic', 'Jørn Utzon', 'Bennelong Point', 'performing arts', 'shell roofs', 'UNESCO World Heritage site', 'Sydney Harbour', 'Danish architect', 'multi-venue', 'ceramic tiles', 'expressionist design'] | ['China', 'France', 'Germany', 'Italy', 'Austria', 'USA', 'Canada', 'Finland', 'Hungary', 'Afghanistan', 'Albania', 'Algeria', 'Greece', 'Argentina', 'Bangladesh', 'Belgium', 'Brazil', 'Cambodia', 'Bulgaria', 'Chile', 'Colombia', 'Croatia', 'Cuba', 'Denmark', 'England', 'Egypt', 'Estonia', 'Ethiopia', 'Iceland', 'India', 'Indonesia', 'Iran', 'Iraq', 'Ireland', 'Israel', 'Jamaica', 'Japan', 'Jordan', 'Kenya', 'Kuwait', 'Lebanon', 'Malaysia', 'Mexico', 'Mongolia', 'Morocco', 'Nepal', 'New Zealand', 'Nigeria', 'Norway', 'Pakistan', 'Peru', 'Philippines', 'Poland', 'Portugal', 'Qatar', 'Romania'] |
| Cristiano Ronaldo | FROM_COUNTRY | is from the country of | Portugal | Fact: Cristiano Ronaldo is from the country of | ['football', 'Real Madrid', 'Manchester United', 'Juventus', 'World Player', 'Euro', 'Nike', 'endorsements', "Ballon d'Or", 'Champions League', 'forward', 'La Liga', 'Serie A', 'free-kicks', 'hat-tricks', 'CR7 brand', 'foundation', 'Museu CR7', 'scoring records'] | ['USA', 'China', 'France', 'Germany', 'England', 'Italy', 'Afghanistan', 'Albania', 'Algeria', 'Argentina', 'Australia', 'Austria', 'Bangladesh', 'Belgium', 'Brazil', 'Bulgaria', 'Cambodia', 'Canada', 'Chile', 'Colombia', 'Croatia', 'Cuba', 'Denmark', 'Egypt', 'Estonia', 'Ethiopia', 'Finland', 'Greece', 'Hungary', 'Iceland', 'India', 'Indonesia', 'Iran', 'Iraq', 'Ireland', 'Israel', 'Jamaica', 'Japan', 'Jordan', 'Kenya', 'Kuwait', 'Lebanon', 'Malaysia', 'Mexico', 'Mongolia', 'Morocco', 'Nepal', 'New Zealand', 'Nigeria', 'Norway', 'Pakistan', 'Peru', 'Philippines', 'Poland', 'Qatar', 'Romania'] |

Table 3: Some full example data points from our dataset, $(s, r, a, S, R)$

## D FURTHER METHODS

Here, we provide details for regarding how to calculate the logit lens, DLA and DLA by source token. We borrow from the notation presented in **?**.

The function a standard transformer with $L$ layers and parameters $\theta$ implements $f_\theta$ can be expressed $f_\theta(x_{\leq t}) = \text{softmax}(\pi_t(x_{\leq t}))$ where $\pi_t$ is a vecotr of logits given by

$$\pi_t = \text{LayerNorm}(z_t^L)W_U$$
$$z_t^l = z_t^{l-1} + a_t^l + m_t^l$$
$$a_t^l = \text{Attn}(z_t^{l-1})$$
$$m_t^l = \text{MLP}(z_t^{l-1}),$$

where LayerNorm() is a LayerNorm normalisation layer, $W_U$ an unembedding matrix, Attn() a multi-head attention layer, and MLP() a two layer perceptron. The dependence on model parameters $\theta$ is left implicit. In common with much of the literature on mechanistic interpretability Elhage et al. (2021), we refer to the series of residual activations $z_i^l$ as the residual stream.

**Logit Lens.** The logit lens (nostalgebraist, 2020) is an interpretability technique for interpreting intermediate activations of language models, through the insights that the residual stream is a linear sum of contributions from each layer (Elhage et al., 2021) and that the map to logits is approximately linear. It pauses model computation early, converting hidden residual stream activations to probability distributions over the vocabulary at each layer.

$$\tilde{\pi}_t^l = \text{LayerNorm}(z_t^l)W_U$$

with $l \leq L$.

**Direct Logit Attribution** (DLA) is an extension of the logit lens technique. It zooms in to individual model components, through the insight that the residual stream of a transformer can be viewed as an accumulated sum of outputs from all model components (Elhage et al., 2021) DLA therefore gives a measure of the direct effect on the of individual model components on model outputs. Mathematically, we may write

$$a_t^l = \text{Attn}(z_t^{l-1}) = \sum_{h=1}^{H} a_h(z_t^{l-1})$$
$$m_t^l = \text{MLP}(z_t^{l-1}) = \sum_{n=1}^{N} m_n(z_t^{l-1}),$$

where we decompose the attention layer into individual attention heads, and the mlp layer into individual neurons (Elhage et al., 2021). DLA corresponds to the sets of logits

$$\tilde{\pi}_t^{l,h} = \text{LayerNorm}(a_h(z_t^{l-1}))W_U$$
$$\tilde{\pi}_t^{l,n} = \text{LayerNorm}(m_n(z_t^{l-1}))W_U$$

**DLA by source token**. We extend this technique for attention heads through the further insight that attention head outputs are a weighted sum of outputs corresponding to distinct attention source position (Elhage et al., 2021), allowing us to quantify how each group of source tokens in turn contributes directly to the logits. To do so note that the attention head contribution with query position $q = l - 1$ is a sum over key (source) positions

$$a_h(z_t^{l-1}) = \sum_{k=1}^{t} \text{attn\_prob}_{l-1,k}\text{LayerNorm}(z_k^{l-1})W_V W_O$$

Unravelling this sum, just as above, gives a separation of attention head DLA contributions by source token.

$$\tilde{\pi}_t^{l,h,k} = \text{LayerNorm}(\text{attn\_prob}_{l-1,k}\text{LayerNorm}(z_k^{l-1})W_V W_O)W_U$$

# E  FURTHER RESULTS

## E.1  MANY ATTRIBUTES ARE EXTRACTED

| Prompt | Attribute | Counterfactual Relation Attributes | Counterfact Subject Attributes |
|---|---|---|---|
| Fact: Tom Brady plays the sport of | football (0) | golf (2), baseball (3), hockey (5) | quarterback (4), NFL (23), Gisele Bündchen (34) |
| Fact: The Eiffel Tower is in the country of | France (0) | Belgium (1), China (13), Germany (14) | Paris (2), Europe (12), Seine River (347) |
| Fact: The Colosseum is in the country of | Italy (0) | Albania (1), Egypt (13), Greece (15) | Rome (2), ancient (33), ruins (97) |
| Fact: England has the capital city of | London (0) | Kuala Lumpur (35), Beijing (40), Dublin (43) | Queen Elizabeth (219), English (236), football (337) |
| Fact: Michael Jordan plays in the league called the | NBA (0) | NFL (9), PGA (13), NHL (34) | United States (6), USA (23), Chicago Bulls (39) |
| Fact: Pasta has the primary macronutrient of | carbohydrates (0) | protein (3), fiber (4), fat (12) | macaroni (49), fettuccine (54), spaghetti (217) |
| Fact: Stephen Hawking is a professor at the university of | Cambridge (0) | Edinburgh (1), Manchester (2), Oxford (3) | CBE (30), England (31), cosmology (46) |
| Fact: Alan Turing is a professor at the university of | Manchester (0) | Cambridge (1), Edinburgh (2), California Institute of Technology (6) | computer science (13), Bletchley Park (29), England (38) |

Table 4: Many attributes are extracted from the sets $S$ and $R$. Rank displayed in brackets. We analyze the rank of many attribute logits, and display the top 3 from each category among those in our dataset. Generally the highest attributes in $R$ have higher rank than the highest in $S$. Sometimes, the highest rank attributes in $S$ are very correlated with $a$ and therefore $r$, e.g. `France` with `Paris`. Often the counterfactual attributes are decorrelated with $r$. For instance `professor at the university of` and `CBE` or `England`. This suggests subject heads 'misfire' and extract these attributes even in contexts that do not necessitate it.

## E.2  OTHER MODELS

In this section, we provide some analogous summary plots to Figures 2 and 3 for the relations `plays the sport of` and `is in the country of` for several other models.

**GPT2-XL (1.5B)**

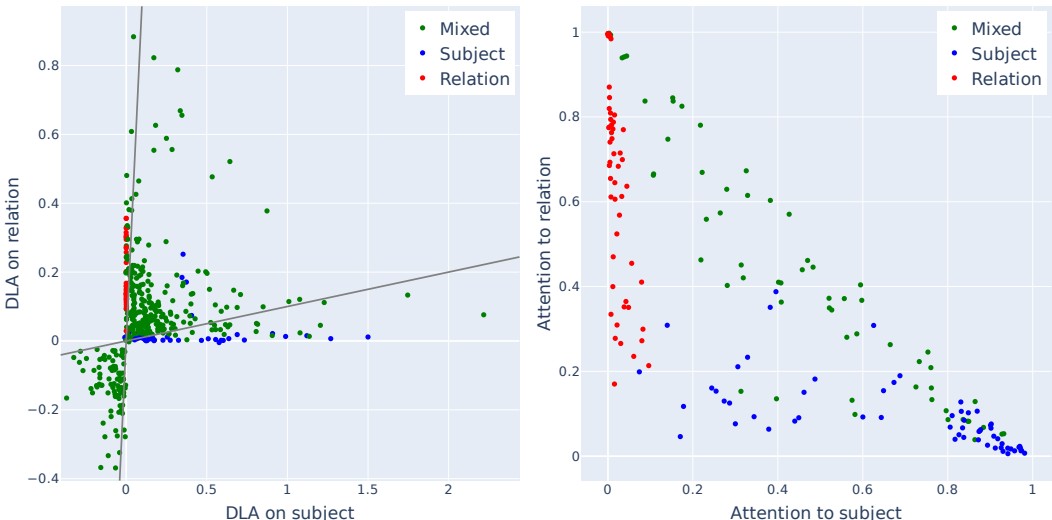

Figure 8: GPT2-XL. Three different types of attention head for factual extraction for prompts of form "$s$ plays the sport of": Subject heads, Relation heads and Mixed heads. (**Left**) DLA on the correct sport, split by attention head *source* token, for top 10 heads by total DLA. Each data point is one prompt for one factual tuple. The gray lines have gradients $1/10$ and $10$ and denote the boundary we use to **define** heads, post averaging, which is somewhat arbitrary. (**Right**) Attention patterns of the top four heads of each kind on each prompt. Subject and Relation heads attend mostly to subjects and relations respectively. Mixed heads attend to both. Attention is not used to define head type.

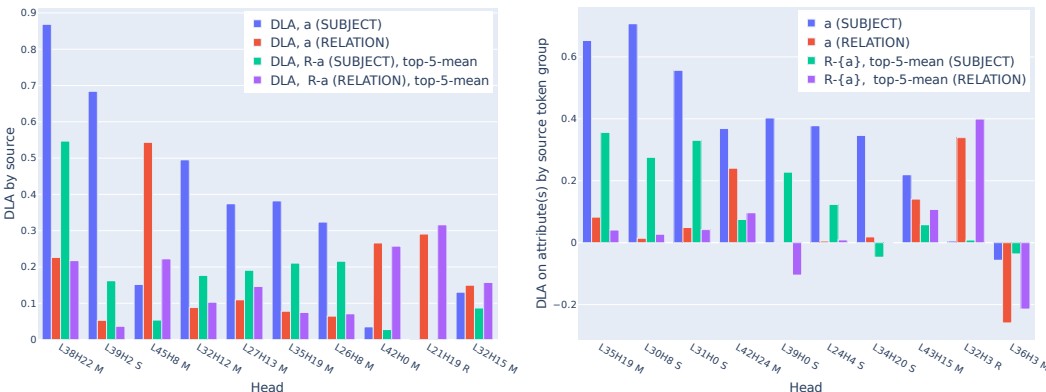

Figure 9: GPT2-XL. Top heads by absolute DLA for the relationships `plays the sport of` (**left**) and for `is in the country of` (**right**), labeled as Subject (S), Relation (R) or Mixed (M). Studying a large set of counterfactual attributes, and splitting by attention source token let's us disentangle these head types. We plot DLA on the attribute $a$, and for the mean of the top 5 attributes in the set $R$ but excluding $a$, both split by attention source token (SUBJECT vs RELATION). All three head types emerge.

**GPT-J (5.6B)**

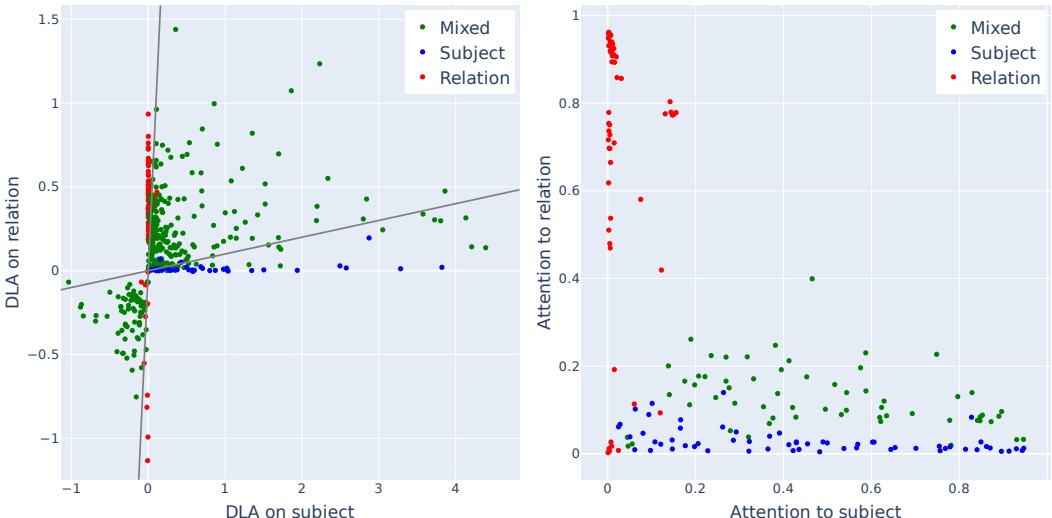

Figure 10: GPT-J. Three different types of attention head for factual extraction for prompts of form "$s$ plays the sport of": Subject heads, Relation heads and Mixed heads. (**Left**) DLA on the correct sport, split by attention head *source* token, for top 10 heads by total DLA. Each data point is one prompt for one factual tuple. The gray lines have gradients $1/10$ and $10$ and denote the boundary we use to **define** heads, post averaging, which is somewhat arbitrary. (**Right**) Attention patterns of the top four heads of each kind on each prompt. Subject and Relation heads attend mostly to subjects and relations respectively. Mixed heads attend to both. Attention is not used to define head type.

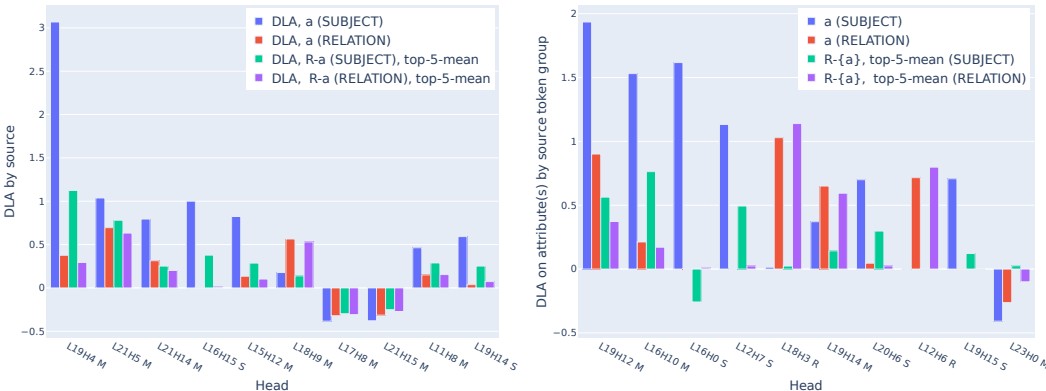

Figure 11: GPT-J. Top heads by absolute DLA for the relationships `plays the sport of` (**left**) and for `is in the country of` (**right**), labeled as Subject (S), Relation (R) or Mixed (M). Studying a large set of counterfactual attributes, and splitting by attention source token let's us disentangle these head types. We plot DLA on the attribute $a$, and for the mean of the top 5 attributes in the set $R$ but excluding $a$, both split by attention source token (SUBJECT vs RELATION). All three head types emerge.

**Pythia-6.9b**

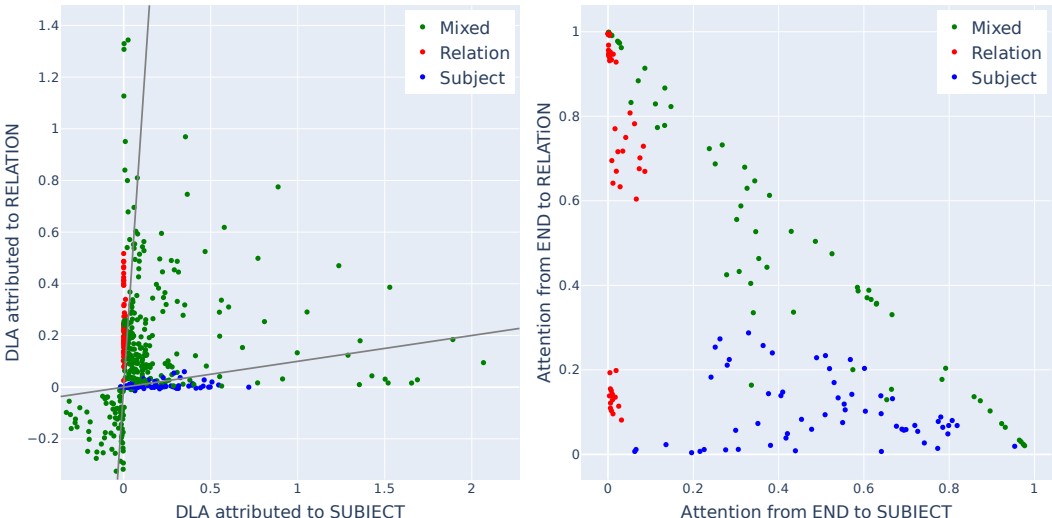

Figure 12: Pythia-6.9b. Three different types of attention head for factual extraction for prompts of form "*s* plays the sport of": Subject heads, Relation heads and Mixed heads. (**Left**) DLA on the correct sport, split by attention head *source* token, for top 10 heads by total DLA. Each data point is one prompt for one factual tuple. The gray lines have gradients $1/10$ and $10$ and denote the boundary we use to **define** heads, post averaging, which is somewhat arbitrary. (**Right**) Attention patterns of the top four heads of each kind on each prompt. Subject and Relation heads attend mostly to subjects and relations respectively. Mixed heads attend to both. Attention is not used to define head type.

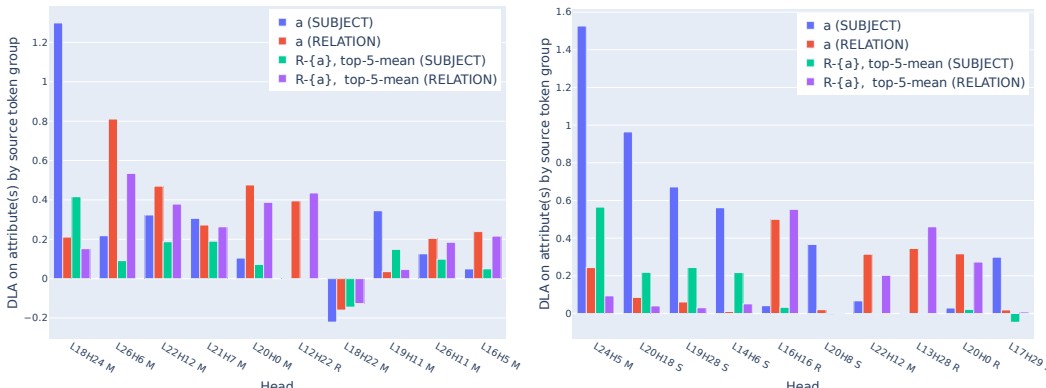

Figure 13: Pythia-6.9b. Top heads by absolute DLA for the relationships `plays the sport of` (**left**) and for `is in the country of` (**right**), labeled as Subject (S), Relation (R) or Mixed (M). Studying a large set of counterfactual attributes, and splitting by attention source token let's us disentangle these head types. We plot DLA on the attribute $a$, and for the mean of the top 5 attributes in the set $R$ but excluding $a$, both split by attention source token (SUBJECT vs RELATION). All three head types emerge.

### E.3 RELATIVE MECHANISM IMPORTANCE

We consider here several measures of importance among the various mechanisms.

**Fraction of heads in each class**. The fraction of heads in each of (subject/relation/mixed) is one possible measure of importance. This varies depending on the choice of subject and relation. See Figure 3 for two examples. There, we see the split of (subject, relation, mixed) among the top 10 heads for the relation "plays the sport of" is (2, 1, 7), but for "is in the country of" it is (4, 2, 4). Inspecting the top 10 heads by DLA for each example in the entire dataset we find 37% of heads get

| relation | baseline loss | subject percent change | relation percent change | mixed percent change loss | mlp percent change |
|---|---|---|---|---|---|
| PLAYS_SPORT | 0.68 | 16.71 | 254.31 | 345.56 | 483.11 |
| IN_COUNTRY | 0.51 | 250.29 | 710.23 | 375.23 | 207.43 |
| CAPITAL_CITY | 1.41 | 67.38 | 206.48 | 90.53 | 222.64 |
| LEAGUE_CALLED | 1.58 | 170.37 | 97.31 | 185.97 | 104.80 |
| PROFESSOR_AT | 0.62 | 127.90 | 503.27 | 78.52 | 714.40 |
| PRIMARY_MACRO | 1.60 | 190.55 | 10.92 | 69.08 | 189.48 |
| PRODUCT_BY | 0.75 | 112.21 | 578.38 | 145.64 | 249.04 |
| FROM_COUNTRY | 1.16 | 196.88 | 234.61 | 101.40 | 255.63 |

Table 5: Percent change in loss when ablating the direct path to logits of each component.

categorised as subject heads and 33% as relation heads, with the remaining 30% as mixed heads. This indicates all three head types are important for the task.

**Logit contribution**. The contribution to logits is another possible choice of metric. Figure 2 visualises this – we can qualitatively see that all three head types are important. We may analyse the percentage of the final (mean centered) logit contributed from each component type, across the entire dataset. We omitted several negatively suppressive components for the purpose of this analysis. Again, we see that the contributions from each type of mechanism is important. Subject heads contribute 18%, relation heads 24%, mixed heads 27%, and the mlp layers 30%, across the entire dataset.

**Ablations**. Naive ablations have been noted in prior work to be counteracted by self-repair in the factual recall set up, a phenomenon known as the hydra effect (?). We therefore followed the approach of (Wang et al., 2022), and performed edge patching - ablating the direct path term between model components and logits. We present in Table 5 baseline loss, together with the loss after knocking out one of the four model mechanisms. Each loss reported is aggregated over the relation dataset. We see knocking out any individual component significantly harms loss in each case.

## E.4 SUBJECT HEADS

| subject | L21H9 PLAYS SPORT | L16H20 PLAYS SPORT | L22H17 PLAYS SPORT | L17H2 IN COUNTRY | L16H12 IN COUNTRY | L17H17 IN COUNTRY | L18H14 PROFESSOR AT |
|---|---|---|---|---|---|---|---|
| Michael Jordan | basketball, shooting, shoot, Basketball, shot, Shot, shoots | Basketball, basketball, NBA | basketball, Basketball, NBA, basket, ho, asketball | USA, US, America, American, USA, Chicago, Americans | | Jordan, Jordan, ordan, Nile | Chicago, Chicago, Illinois |
| David Beckham | Soccer, soccer, football, Football, Football, footballers, Soc | Soccer, soccer, FIFA, MLS | | London, UK, England, British, London, Britain, English | | | |
| Roger Federer | tennis | singles, ATP, tournament, tournaments, tennis | court, final, final, court, courts, Rac, serve | Switzerland, Swiss, global, global | Swiss | | |
| Stephen Hawking | | | | England, Britain, London, British, Brit, England, UK | | | Cambridge, Cambridge, calculation, mathematic |
| Niels Bohr | energies, ATP, energy | | | | Swedish, Sweden, Swed, Å, Danish | r | Philosophy |
| The Colosseum | fight | | | Italy, Rome, Roman, Romans, Italian, Ital, Italian | Italy, Italian, Italian, Rome, Ital, Milan | Italy, Italian, Italian, Rome, Ital, Roma | |
| The Taj Mahal | | | | India, Indian | | Indian, Indian, Indians, Pakistani, India, India, Shah | |
| The Eiffel Tower | | | | Paris, Paris, France, France, London, London | France, Paris | | |

Table 6: Using head $OV$ circuits as probes acting on the enriched subject representation's final token residual stream elicits interpretable attributes in the head category $C$ as the top few DLA tokens. We include the relation for which the head is a subject head for in the column titles. We only include attributes the head is sufficiently confident about ($> 1\%$). For instance, applying the head L21H9 to sports players usually elicits their sport. Applying it to `The Colosseum` elicits `fight`, and with lower confidence `boxing`, which falls within the same category.

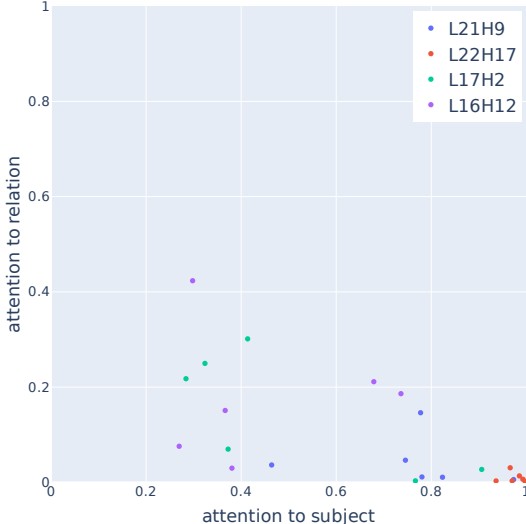

Figure 14: Attention scores of several subject heads on prompts with subject `Michael Jordan` with a range of different relations, pertaining to sport, country, language, etc. We see several interesting attention patterns. L22H17 attends quite uniformly to the subject here, while other subject heads have more variable attention patterns.

In Figure 14 we analyze the attention patterns of several subject heads, across a range of prompts with a single fixed subject $s$, but different relationships $r$. We see significant attention to SUBJECT no matter what prompt is given, i.e. these heads often extract attributes irrelevant to the relationship. We find several kinds of interesting attention pattern. (1) Heads that always attend to the subject with very high probability, independent of the relationship given in the prompt (e.g. Layer 22 Head 17 (L22H17)), for basketball players). This correlates with the attributes this head extracts, only the sport of basketball, no other sports. Notably, this head does not have as high attention on non-basketball sports players. (2a) Heads that pay variable attention to the subject, in a mostly uninterpretable way. (2b) Heads paying variable attention to the subject, in an interpretable fashion, dependent on the prompt (e.g. L17H2, which attends more if the prompt requests a country or city). These latter heads, by virtue of attending from END to SUBJECT can only be influenced by the relation on the query side. This is an instance of query composition Elhage et al. (2021), as suggested by Geva et al. (2023a). We however note this mechanism is relatively unimportant among the studied examples – we only found a handful of instances of this, all of which related to country attributes.

E.5 RELATION HEADS

In this section, we provide further results on relation heads.

### E.5.1 RELATION HEADS PULL OUT MANY ATTRIBUTES IN THE SET R.

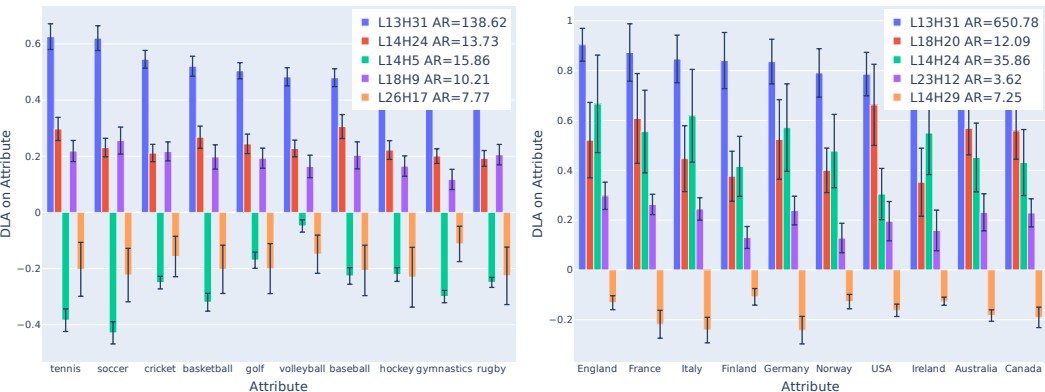

Figure 15: Relationship heads pull out many attributes consistently across a range different subjects for prompts with relationship `plays the sport of` and `is in the country of`. The error bars are standard deviation over subjects, and them beings small suggests that these heads do not meaningfully depend on the subject. We also include the mean attention ratio of to RELATION over SUBJECT for each head.

### E.5.2 RELATION HEADS PRIMARY FUNCTION IS OFTEN TO BOOST ATTRIBUTES IN THE SET R.

| prompt | most important relation head | second most important relation head |
|---|---|---|
| Fact: Michael Jordan plays the sport of | games (0), roles (1), role (2), genres (3), soccer (5), tennis (8), sport (9), Role (10), lite (12), football (13), genre (14), ballet (16), cricket (18), disciplines (21), athlet (23), games (26), violin (27), basketball (30), bass (31), biology (33), sports (34), afers (35), music (37), slots (40), slot (41), battles (42), golf (43), Wrestling (46), volley (49) | football (2), Football (5), chess (8), Wrestling (9), baseball (14), opera (16), football (17), switch (18), JavaScript (19), tennis (22), Football (26), JavaScript (31), guitar (34) |
| Fact: The Eiffel Tower is in the country of | territory (0), territories (1), countries (2), country (3), England (4), Kingdom (5), States (6), Region (7), Netherlands (8), province (9), France (10), region (11), Province (12), regions (13), place (14), Germany (15), provinces (16), Finland (17), Italy (18), Region (19), Norway (20), states (21), America (22), area (23), Britain (24), Spain (25), States (26), Territory (27), USA (28), regions (29), Sweden (30), region (31), Ireland (32), northern (33), country (34), Australia (35), Denmark (36), Canada (37), land (38), Arabia (39), place (41), France (42), England (43), kingdom (44), areas (45), area (46), realm (47), Switzerland (48), homeland (49) | abroad (0), country (1), countries (2), overseas (3), country (4), international (5), internationally (6), international (7), Country (8), expatri (9), foreigners (10), expatriate (11), foreign (12), France (13), national (15), national (16), France (17), país (18), nationality (19), Country (20), nation (21), nations (22), export (23), Germany (24), foreign (25), nationals (26), Germany (27), países (28), passport (29), exported (30), International (31), visa (32), England (33), International (34), USA (35), embassy (36), Foreign (37), visas (38), USA (39), travel (40), Foreign (41), Switzerland (42), England (43), extrad (44), UK (45), Europe (47), travel (48), Belgium (49) |
| Fact: England has the capital city of | city (0), cities (1), city (2), City (3), City (4), metropolitan (5), urban (6), London (7), Cities (8), street (9), CITY (10), London (11), streets (12), Mayor (13), municipal (14), NYC (15), street (16), downtown (17), urban (18), Metropolitan (19), metro (20), mayor (21), Municipal (22), Street (23), Paris (24), nationwide (25), Urban (26), borough (27), ciudad (28), Paris (29), Delhi (30), town (31), Metro (32), hometown (33), Dublin (34), suburbs (35), overseas (36), regional (37), Mumbai (38), Street (39), Downtown (40), residents (41), Amsterdam (42), Philadelphia (43), capital (44), Chicago (45), Edinburgh (46), abroad (47), national (48), Madrid (49) | cities (0), towns (1), Cities (2), city (3), town (4), hometown (5), city (6), municipalities (7), town (8), City (9), CITY (10), City (11), locations (12), metropolitan (13), Town (14), villages (15), Town (16), locations (17), ville (18), location (19), location (20), stown (21), centres (22), places (23), Sites (24), London (25), destinations (27), headquarters (28), neighborhoods (29), capital (30), localities (31), metro (32), London (33), place (34), centers (35), sites (36), downtown (37), sites (38), Location (39), located (40), Place (41), regions (42), counties (43), venues (44), ports (45), develop (46), ciudad (47), ville (48), apolis (49) |
| Fact: Michael Jordan plays in the league called the | bas (0), Draft (1), draft (4), NBA (6), fil (8), drafting (9), bas (10), draft (11), drafted (12), offseason (15), fil (16), MLB (29), NHL (32), preseason (36), Steelers (41), cent (42), (49) | player (0), players (1), league (2), championship (3), stadium (4), club (5), NFL (6), franchise (7), team (8), player (9), teams (10), clubs (11), NBA (12), fans (13), coaches (14), football (15), game (16), coach (17), soccer (18), coaching (19), Players (20), teammates (21), franch (22), leagues (23), squad (24), athletes (25), referee (26), training (27), athlete (28), games (29), hockey (30), tournament (31), basketball (32), team (33), Stadium (34), championships (35), rookie (36), Championship (37), baseball (38), ESPN (39), injury (40), preseason (41), club (42), competitive (43), roster (44), season (45), NCAA (46), teammate (47), Player (48), |
| Fact: Stephen Hawking is a professor at the university of | University (0), university (1), universities (2), University (3), College (4), UK (5), Academic (6), UK (8), college (9), colleges (10), College (11), England (12), institute (13), 's (14), Univers (15), ' (16), Institute (17), itself (18), British (19), Univers (20), UCLA (21), Cambridge (22), Faculty (23), institution (25), Zealand (26), undergraduate (27), Britain (28), Academy (29), (30), academy (31), Yale (32), academic (34), England (35), Cambridge (36), overrightarrow (39), Harvard (40), Enum (41), Oxford (42), achus (43), professors (44), Ireland (45), School (47), Scotland (48), $' (49) | |
| Fact: Chicken has the primary macronutrient of | nutrients (0), nutrient (1), nutrition (2), dietary (3), vitamins (4), nutritional (5), sugars (6), carbohydrates (7), glucose (8), protein (9), carbohydrate (10), energy (11), proteins (12), calories (13), iron (14), minerals (15), amino (16), vitamin (17), metabolic (18), metabolism (19), Nutrition (20), Diet (21), Protein (22), diet (23), diets (24), nutrients (25), lipids (26), Vitamin (27), Energy (28), fatty (29), sugar (30), insulin (31), calcium (32), energy (33), lipid (34), Energy (35), protein (36), fats (37), nutrition (38), metabol (39), phosphorus (40), Proteins (41), Iron (42), carot (43), Protein (44), glucose (45), iron (46), selenium (47), collagen (48), fat (49) | Quantity (0), iv (1), olean (2), carbon (3), leen (6), Judaism (7), rice (9), organic (15), beef (25),electrons (42), |

Table 7: Some of the top few DLA tokens for the top two relation heads corresponding to a range of relations. Manually sampled relevant words from the top 50 output tokens, together with rank in brackets. There are many interesting things to note. For example, the top relation head for `plays the sport of` extracts both sports, as well as other things one can `play` - the category $C$ of this head is wider than just sports.

### E.5.3 RELATION HEADS (MOSTLY) DO NOT HAVE SIGNIFICANT INDIRECT EFFECT DEPENDENT ON THE SUBJECT.

We perform patching experiments where we patch the subject between two prompts with the same relationship on the top 5 relation head outputs, and measure the difference in performance. We find that for some relationships, performance is invariant. If the relation heads causally depended on specific features of the subject, we would expect to see a large decrease in performance.

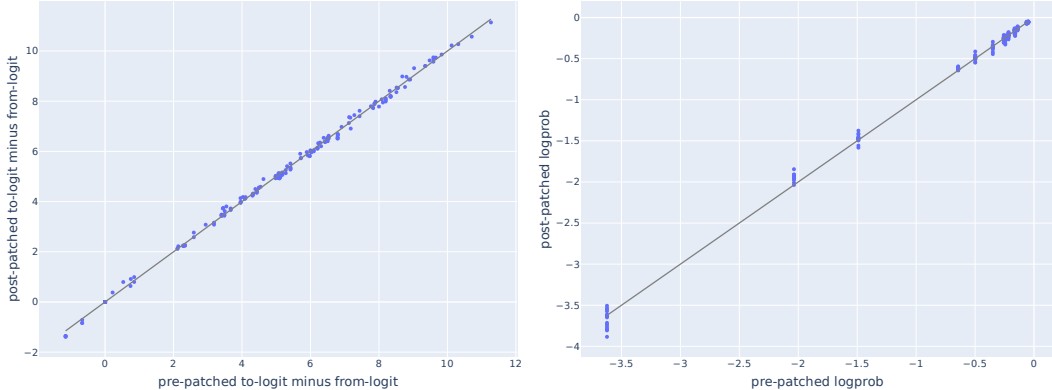

Figure 16: We patch the top 5 relation heads outputs for prompts with relation `plays the sport of` between different subjects to study the indirect effect of relation heads. For this relationship, we see that on average, performance does not increase for both a logit-diff between to-logit and from-logit (left) and logprob based (right) metric. The gray line indicates no change.

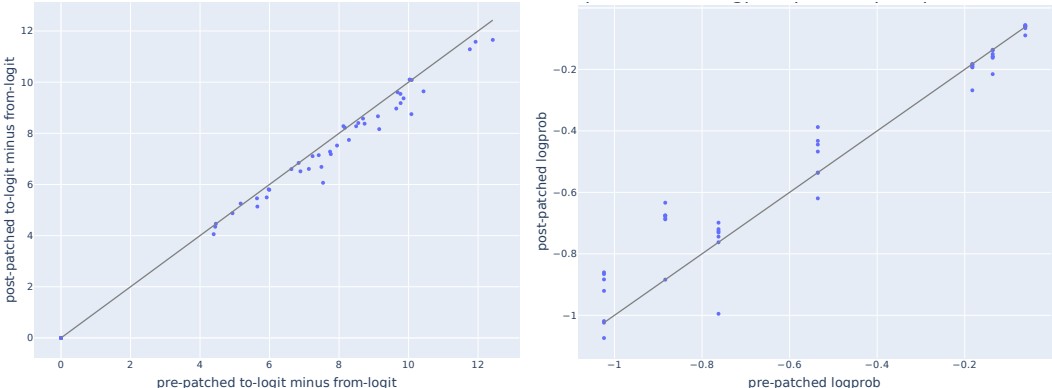

Figure 17: We patch the top 5 relation heads outputs for prompts of form `is in the country of` between different subjects to study the indirect effect of relation heads. We see that on average, logprob does not decrease, but logit difference between the to-logit and from-logit decrease slightly. We generally see that patching improves performance on low probability outputs. This suggests the model has some 'confidence' feature that gets modified through patching. The gray line indicates no change.

### E.5.4  SUBJECT-RELATION PROPAGATION

In Pythia-2.8b, we found that relation heads generally did not privilege the correct attribute $a$ among the set $R$. When investigating a larger Pythia 6.9B model, we observed relation heads frequently extract the correct attribute whilst attending *only* to RELATION, for a variety of different subject/attributes. For example, with `s plays the sport of` prompts, we found attention head L26H6 can extract `basketball` when given `Michael Jordan` as the subject, and `soccer` when given `David Beckham` as the subject.

We hypothesize that there are two mechanisms here. Firstly some subject head attends from `sport` to the subject, and propagates facts (including the sport and other facts about the subject). Then the relationship head L26H6 receives both a large number of sports from the usual mechanism, but also a boosted correct sport that was already moved to the same place in the `sport` residual stream.

We verified this hypothesis through 'attention-knockout', zero-ablating all attention from RELATION to SUBJECT. This resulted in head L26H6 instead of extracting a consistent set of sports regardless of the player, and not privileging the correct attribute $a$. This head remains the most important relation head by DLA for a variety of `plays the sport of` prompts.

The general takeaway from this finding is that our set of mechanisms, may not be completely universal (Olah et al., 2020) across model scale, and we should expect larger models to implement more sophisticated circuits.

### E.6 MIXED HEADS

#### E.6.1 INSPECTING SORTED DLA

To illustrate the facts extracted by a selection of mixed heads and prompts, we investigate DLA by source token, for all vocab tokens.

We note that for some heads $C \approx R$, i.e. the head's category of specialization is similar to the relationship $r$ that is being investigated. For example, we show in Table 11 that L22H15 is specialized to the categories of sport and communication, which overlaps significantly with the `plays the sport of` prompts. Similarly L23H22 was found to be a countries and languages extractor, overlapping significantly with the `is in the country of` prompts. With these heads, the correct attribute is consistently one of the top tokens from SUBJECT, and high but not top from RELATION.

In many cases there is less overlap between $C$ and $R$. For example L17H30 appears to specialize in "players/things that can be played". This head does have the correct attribute for `plays the sport of` prompts in the top tokens for the subject, but it gives a higher DLA for more generic terms that also could reasonably fit within $C \cap S$ and $C \cap R$ (e.g. `player` and `team`). Understanding the category of head specialization is therefore useful in interpreting this type of mixed head.

| head | prompt | subject | relation | total |
|---|---|---|---|---|
| L22H15 | Fact: Michael Jordan plays the sport of | **0. basketball (0.397)** 
 1. football (0.354) 
 2. sports (0.344) 
 3. soccer (0.336) 
 4. footballers (0.327) | 0. games (1.082) 
 **1. basketball (1.026)** 
 2. sports (0.990) 
 3. game (0.987) 
 4. sport (0.948) | **0. basketball (1.427)** 
 1. games (1.354) 
 2. sports (1.340) 
 3. game (1.267) 
 4. sport (1.264) |
| L22H15 | Fact: Mike Trout plays the sport of | **0. baseball (0.522)** 
 1. Baseball (0.447) 
 2. MLB (0.434) 
 3. teammates (0.401) 
 4. sports (0.388) | 0. games (0.610) 
 **1. baseball (0.603)** 
 2. players (0.602) 
 3. Players (0.597) 
 4. player (0.593) | **0. baseball (1.127)** 
 1. players (0.992) 
 2. sports (0.988) 
 3. Players (0.971) 
 4. player (0.957) |
| L22H15 | Fact: Tom Brady plays the sport of | **0. football (0.893)** 
 1. Football (0.810) 
 2. NFL (0.806) 
 3. Football (0.792) 
 4. football (0.792) | **0. football (0.561)** 
 1. games (0.546) 
 2. players (0.522) 
 3. player (0.518) 
 4. Football (0.514) | **0. football (1.456)** 
 1. Football (1.313) 
 2. Football (1.305) 
 3. football (1.292) 
 4. NFL (1.198) |
| L17H30 | Fact: Michael Jordan plays the sport of | 0. games (0.355) 
 1. players (0.352) 
 2. player (0.338) 
 ... 
 **23. basketball (0.233)** | 0. players (1.371) 
 1. player (1.330) 
 2. play (1.315) 
 ... 
 **43. basketball (0.550)** | 0. players (1.716) 
 1. player (1.663) 
 2. play (1.596) 
 ... 
 **34. basketball (0.782)** |
| L17H30 | Fact: Mike Trout plays the sport of | 0. players (0.229) 
 1. player (0.216) 
 2. teams (0.187) 
 3. games (0.185) 
 **4. baseball (0.179)** | 0. players (1.300) 
 1. player (1.260) 
 2. play (1.200) 
 ... 
 **55. baseball (0.488)** | 0. players (1.513) 
 1. player (1.463) 
 2. play (1.333) 
 ... 
 **42. baseball (0.661)** |
| L17H30 | Fact: Tom Brady plays the sport of | 0. players (0.265) 
 1. player (0.247) 
 2. Players (0.241) 
 ... 
 **10. football (0.214)** | 0. players (1.428) 
 1. player (1.397) 
 2. play (1.365) 
 ... 
 **31. football (0.692)** | 0. players (1.684) 
 1. player (1.638) 
 2. play (1.591) 
 ... 
 **28. football (0.902)** |
| L18H25 | Fact: Michael Jordan plays the sport of | 0. skating (0.246) 
 1. skate (0.210) 
 2. Stadium (0.182) 
 ... 
 **9. basketball (0.151)** | 0. sport (0.198) 
 1. Sport (0.192) 
 2. tennis (0.184) 
 ... 
 **58. basketball (0.107)** | 0. skating (0.405) 
 1. skate (0.354) 
 2. sport (0.331) 
 ... 
 **19. basketball (0.257)** |
| L18H25 | Fact: Mike Trout plays the sport of | 0. Golf (0.045) 
 1. leaf (0.038) 
 2. golf (0.037) 
 ... 
 **274. baseball (0.016)** | 0. Sport (0.081) 
 1. sport (0.080) 
 2. skiing (0.077) 
 ... 
 **129. baseball (0.036)** | 0. Golf (0.097) 
 1. Track (0.093) 
 2. golf (0.093) 
 ... 
 **29. baseball (0.062)** |
| L18H25 | Fact: Tom Brady plays the sport of | 0. Formula (0.009) 
 1. luggage (0.008) 
 2. Stadium (0.008) 
 ... 
 **646. football (0.004)** | 0. Sport (0.333) 
 1. sport (0.327) 
 2. skiing (0.308) 
 ... 
 **102. football (0.160)** | 0. Sport (0.323) 
 1. sport (0.315) 
 2. sports (0.304) 
 ... 
 **70. football (0.168)** |
| L23H22 | Fact: The Colosseum is in the country of | **0. Italy (0.960)** 
 1. Italian (0.954) 
 2. Italian (0.914) 
 3. Ital (0.860) 
 4. Rome (0.722) | **0. Italy (0.304)** 
 1. Italian (0.232) 
 2. Ital (0.222) 
 3. Rome (0.220) 
 4. Italian (0.216) | **0. Italy (1.257)** 
 1. Italian (1.179) 
 2. Italian (1.125) 
 3. Ital (1.076) 
 4. Rome (0.938) |
| L23H22 | Fact: The Eiffel Tower is in the country of | 0. French (1.229) 
 **1. France (1.176)** 
 2. French (1.111) 
 3. Paris (1.081) 
 4. France (1.031) | **0. France (0.416)** 
 1. France (0.364) 
 2. Paris (0.347) 
 3. French (0.305) 
 4. Paris (0.300) | **0. France (1.589)** 
 1. French (1.531) 
 2. Paris (1.427) 
 3. France (1.394) 
 4. French (1.371) |
| L23H22 | Fact: The Taj Mahal is in the country of | **0. India (0.863)** 
 1. India (0.795) 
 2. Indian (0.734) 
 3. Pakistan (0.684) 
 4. Indian (0.645) | **0. India (0.248)** 
 1. Pakistan (0.223) 
 2. India (0.219) 
 3. istan (0.199) 
 4. Arabia (0.198) | **0. India (1.106)** 
 1. India (1.010) 
 2. Pakistan (0.907) 
 3. Indian (0.851) 
 4. Indian (0.737) |
| L26H8 | Fact: The Colosseum is in the country of | 0. Rome (1.110) 
 **1. Italy (0.963)** 
 2. Italian (0.962) 
 3. Italian (0.918) 
 4. Ital (0.911) | 0. Italian (0.109) 
 **1. Italy (0.109)** 
 2. Italian (0.106) 
 3. Ital (0.099) 
 4. Rome (0.095) | 0. Rome (1.206) 
 **1. Italy (1.072)** 
 2. Italian (1.071) 
 3. Italian (1.024) 
 4. Ital (1.010) |
| L26H8 | Fact: The Eiffel Tower is in the country of | 0. French (1.953) 
 1. Paris (1.952) 
 2. Paris (1.845) 
 3. French (1.829) 
 **4. France (1.815)** | 0. French (0.095) 
 1. French (0.090) 
 **2. France (0.086)** 
 3. Paris (0.086) 
 4. Paris (0.081) | 0. French (2.048) 
 1. Paris (2.038) 
 2. Paris (1.926) 
 3. French (1.919) 
 **4. France (1.901)** |
| L26H8 | Fact: The Taj Mahal is in the country of | 0. Paris (0.211) 
 1. French (0.205) 
 2. Paris (0.198) 
 3. French (0.194) 
 4. France (0.190) | 0. Spanish (0.007) 
 1. Spain (0.007) 
 2. Spain (0.007) 
 3. Barcelona (0.007) 
 4. Portuguese (0.007) | 0. Paris (0.208) 
 1. Paris (0.196) 
 2. French (0.194) 
 3. France (0.191) 
 4. French (0.189) |

Table 8: Sorted DLA over all vocab tokens, broken down by source tokens (SUBJECT or RELATION). We note that for mixed heads where $C \approx R$ such as L22H15 (a head with a specialized category of sports), the correct attribute is consistently one of the top tokens from SUBJECT, and high but not top from RELATION. By contrast, for mixed heads with slightly different specializations, the correct attribute is high but not top from both SUBJECT and RELATION.

### E.6.2 SUBJECT-RELATION PROPAGATION WITH MIXED HEADS

In order to provide a clear distinction between the attributes extracted in the SUBJECT and RELATION tokens, we also investigated knocking out attention from all baring the last RELATION tokens to SUBJECT. This prevents the correct attribute from having already been moved into earlier RELATION tokens.

We find that the DLA from the relation tokens increases significantly, which demonstrates that some information about the subject had already propagated to earlier RELATION tokens. By isolating this effect through attention knockout, we confirm that mixed heads where $C \approx R$ regularly result in the attribute being the top token from SUBJECT and near, but not at, the top from RELATION.

| Head | Prompt | Subject | Relation (Without Knockout) | Relation (With Knockout) | Relation Change |
|------|--------|---------|------------------------------|---------------------------|-----------------|
| L22H15 | Fact: Michael Jordan plays the sport of | 0 | 1 | 0 | -1 |
| L22H15 | Fact: Mike Trout plays the sport of | 0 | 1 | 3 | +2 |
| L22H15 | Fact: Tom Brady plays the sport of | 0 | 0 | 9 | +9 |
| L17H30 | Fact: Michael Jordan plays the sport of | 23 | 43 | 52 | +9 |
| L17H30 | Fact: Mike Trout plays the sport of | 4 | 55 | 139 | +84 |
| L17H30 | Fact: Tom Brady plays the sport of | 7 | 31 | 37 | +6 |
| L18H25 | Fact: Michael Jordan plays the sport of | 9 | 58 | 60 | +2 |
| L18H25 | Fact: Mike Trout plays the sport of | 277 | 129 | 84 | -45 |
| L18H25 | Fact: Tom Brady plays the sport of | 550 | 102 | 93 | -9 |
| L23H22 | Fact: The Colosseum is in the country of | 0 | 0 | 15 | +15 |
| L23H22 | Fact: The Eiffel Tower is in the country of | 1 | 0 | 0 | 0 |
| L23H22 | Fact: The Taj Mahal is in the country of | 0 | 0 | 0 | 0 |
| L26H8 | Fact: The Colosseum is in the country of | 1 | 1 | 3 | +2 |
| L26H8 | Fact: The Eiffel Tower is in the country of | 4 | 2 | 2 | 0 |
| L26H8 | Fact: The Taj Mahal is in the country of | 22630 | 22594 | 25454 | +2860 |
| L21H23 | Fact: The Colosseum is in the country of | 62 | 0 | 77 | +77 |
| L21H23 | Fact: The Eiffel Tower is in the country of | 33 | 2 | 4 | +2 |
| L21H23 | Fact: The Taj Mahal is in the country of | 1 | 2 | 2 | 0 |
| Mean | | 1311 | 1278 | 1446 | +167 |

Table 9: The rank of the correct attribute from RELATION increases when we knock out attention from earlier RELATION tokens to SUBJECT. This suggests significant subject-relation propagation otherwise occurs of the correct fact.

### E.7 MLPS

#### E.7.1 ATTRIBUTES IN $R$ ARE CONSISTENTLY BOOSTED BY MLPS

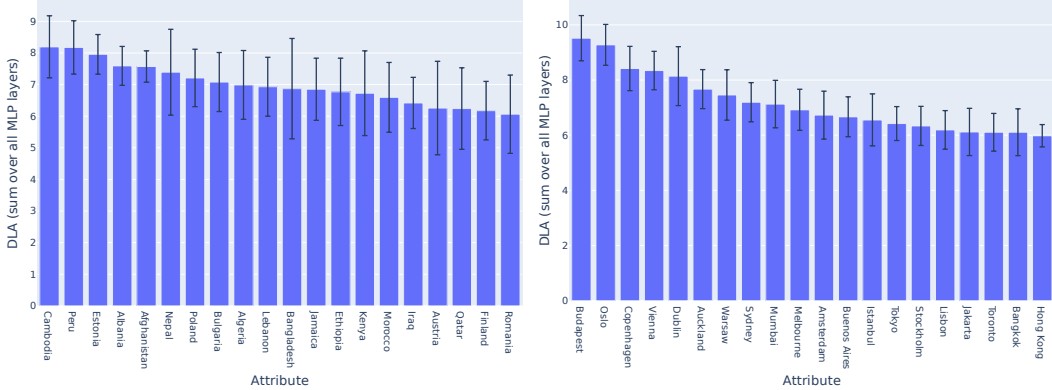

Figure 18: Many attributes in $R$ are boosted by MLPs over a range of prompts with relations `in the country of` (**left**) and `has capital city` (**right**), independent of which subject is given. Error bars are standard deviation over different subjects, which are small. This suggests the direct effect of the MLP does not causally depend on the subject.

### E.7.2 The primary direct effect of MLPs is often to boost many attributes in the set $R$.

| Prompt | Top MLP Logit Lens Tokens |
|---|---|
| Fact: Michael Jordan plays the sport of | Wrest (0), squash (1), skiing (4), lineback (11), surfing (12), rugby (13), Rugby (14), volley (15), Running (17), boxing (21), Baseball (22), cricket (27), Floor (28), cycling (29), shooting (30), Mixed (31), gardening (32), Golf (34), Forward (35) swimming (39), bridge (40), coward (42), paddle (43), impat (44), CLUDE (45), ping (46), escaping (47), contacting (48), flag (49) |
| Fact: The Eiffel Tower is in the country of | Niger (0), Burk (1), Georgia (2), Aust (3), Eston (4), Zimbabwe (5), Utt (6), Gren (7), Trin (8), Haiti (9), Lithuan (10), Guatemala (11), Lub (12), Hond (13), Liber (14), Equ (15), Bangladesh (16), Yug (17), Tun (18), Ly (19), Belf (20), Myanmar (21), Kenya (22), Hawai (23), Nepal (24), Sen (25), Ecuador (26), Yemen (27), Iraq (28), Cambodia (29), Chin (30), Afghanistan (31), Turk (32), Chad (33), Somalia (34), Alaska (35), Continuous (36), Tanzania (37), Sloven (38), Peru (39), Idaho (40), Bul (41), Aqu (42), Albany (43), fered (44), Norfolk (45), Byz (46), Kazakh (47), Tuc (48), Bulgaria (49) |
| Fact: Stephen Hawking is a professor at the university of | Adelaide (0), Cape (5), Alaska (6), Cinc (7), Cincinnati (8), Hawai (12), Manchester (13), Manit (21), Cam (22), fered (23), Chester (24), Chel (25), Gib (26), icago (28), Manila (29), Sussex (31), Minn (33), Buenos (40), Ald (41), Ald (42), Malta (45), Calgary (46), Leicester (48) |
| Fact: England has the capital city of | Budapest (0), Oslo (1), Birmingham (2), Belfast (3), Cincinnati (4), Constantin (5), Sask (6), Manchester (7), Lancaster (8), Kingston (9), Vienna (10), Malta (11), Copenhagen (12), Guatemala (13), Byz (14), Fuk (15), Chester (16), Brighton (17), Ottawa (18), Trin (19), Helsinki (20), Sacramento (21), Adelaide (22), Omaha (23), Winnipeg (24), Lah (25), Newcastle (26), Mumbai (27), Concord (28), Manila (29), Prague (30), Warsaw (31), Newport (32), Lans (33), Hartford (34), Rochester (35), Glasgow (36), Bulgaria (37), Card (38), Pret (39), Derby (40), Richmond (41), Windsor (42), Buenos (43), Calgary (44), Leeds (45), Dublin (46), Tun (47), Lok (48), Hull (49), Jak (50) |

Table 10: Top DLA tokens on the sum of all MLP layers tend to be attributes in the set $R$. Rank is included in brackets. Often, they are attributes we did not pragmatically check through inclusion in our dataset sets $S$ and $R$. For instance, our set $R$ for `is in the country of` did not include the country of `Burkina Faso`, which is the rank 1 attribute pushed for by the MLP for the prompt `The Eiffel Tower is in the country of`. The correct attribute $a$ is not privileged among these, and is often quite low in rank.

**MLPs on END (mostly) do not have significant indirect effect dependent on the subject.**

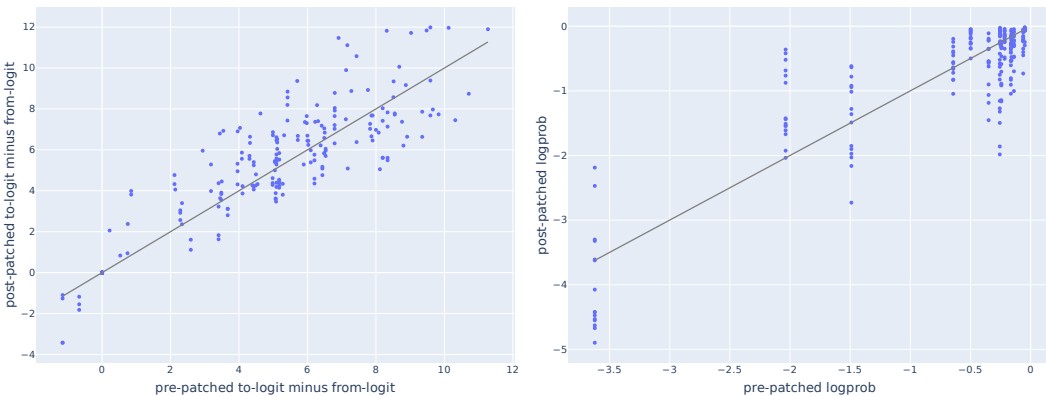

Figure 19: We patch the all MLP outputs on END for prompts of form `plays the sport of` between different subjects to study the indirect effect of MLPS on the END token. For this relationship, we see that on average, performance does not increase for both a logit-diff between to-logit and from-logit (left) and logprob based (right) metric. The grey line indicates no change.

Note that, for some relations, the MLP *does* have significant indirect effect. We do not explain these cases, instead opting to only explain *part* of the function of the MLP through it's direct effect.

### E.8 Category Identification

Here we try to better understand head categories, by inspecting the top head DLA tokens on a wider distribution of factual recall prompts. We looked at top DLA tokens extracted from 10,000 randomly selected prompts from the `CounterFact` dataset (Meng et al., 2023a). A summary of some of these

are included below for the top 3 Subject, Relation and Mixed heads. We find that head categories are not quite aligned with $S$ or $R$ – heads are **polysemantic** (Elhage et al., 2022).

In Section 3.2, we saw the relation head L13H31 responded to both sports an countries. It may do this because it appears to be specialized to locations/position/places, and most sports are also the start of places (e.g. `basketball` can be the sport or the first token in `basketball stadium`.

We also note that some heads misfire, extracting irrelevant attributes, as well as relevant ones. L18H25 appears to be specialized to the category of transport and consumables. However, this head is the 8th most important mixed head across the `plays the sport of` and `is in the country of` prompts (by DLA). Upon investigation we believe this is due to there being some cross-over between sports and transport words, such as `Golf` (a car brand and also a sport), `swimming` (a means of traveling and also a sport) and `track` (a railway track and also the sport of track and field).

| Head | Type | Theorized Categories | Top 50 Tokens |
|---|---|---|---|
| L13H31 | Relation | Location, positioning, places | locations, location, places, cities, place, towns, locate, sites, languages, positions, located, spots, position, hometown, spot, locating, where, continents, loc, placement, territory, professions, city, states, site, town, headquarters, anywhere, kingdom, countries, municipalities, metropolitan, wherever, roles, regions, country, territories, vicinity, camps, venues, venue, centers, placed, destinations, france, residence, placing, finland, island, positioning |
| L14H24 | Relation | Location, direction, languages | locations, location, anywhere, places, loc, located, wherever, somewhere, downtown, place, placement, english, regions, nearby, east, everywhere, vicinity, positions, geographical, localization, north, zones, situated, languages, nearest, southeast, locating, localities, sites, geographic, northeast, elsewhere, placed, south, locate, northwest, language, proximity, locality, geography, locale, nearer, point, spots, outside, areas, travels, hebrew, centralized, centers |
| L17H2 | Subject | International relations, politics | france, french, paris, european, international, europeans, europe, german, germany, public, germans, london, global, eu, england, uk, british, translated, franc, fran, worldwide, britain, monsieur, eur, europ, us, translations, globally, translation, internationally, euro, franois, brit, translate, translator, russian, europa, europea, deutsch, russia, montreal, philippe, publicly, canada, russians, translating, canadian, berlin, jacques, english |
| L17H17 | Subject | Countries, ethnicities, politicians | arizona, alabama, ariz, tamil, indian, kerala, india, nigeria, japanese, pakistan, az, nigerian, japan, ala, pakistani, phoenix, seoul, poland, greek, ari, indians, italy, polish, tokyo, istanbul, delhi, athens, birmingham, punjab, cyprus, greece, turkish, italian, turkey, hindu, niger, venice, lebanese, hawaiian, tampa, warsaw, turk, sic, hawaii, pak, ital, greeks, mumbai, abama, tuc |
| L18H20 | Relation | Places, diplomacy | countries, city, country, nations, international, globally, diplomatic, abroad, worldwide, global, diplomats, ads, europe, internationally, nation, governor, continents, campus, france, legislators, treaty, street, legislative, foreigners, diplomacy, cities, european, overseas, ticket, national, expatri, diplomat, attendees, pases, foreign, capitol, germany, delegates, asia, conference, nationals, student, expatriate, globe, americas, downtown, students, eur, faculty, australia |
| L21H9 | Subject | Hedonism, wealth, sport, violence | stock, wrest, beer, tennis, wrestling, coffee, gun, beers, brewery, soccer, brew, tenn, drink, atp, stocks, football, drunk, guns, fighters, drank, fighter, drinking, alcohol, shooting, footballers, drinks, golf, firearm, drunken, shoot, wwe, fight, fifa, alcoholic, beverage, brewing, play, firearms, bullets, nra, vince, caffeine, shooter, mma, starbucks, fighting, train, beverages, shot, liquor |
| L22H15 | Mixed | Communication, sports | television, tv, games, football, soccer, broadcast, game, sports, broadcasting, players, sport, fifa, player, broadcasts, basketball, radio, payment, hockey, baseball, footballers, tournament, tennis, league, tele, sporting, rugby, gamers, espn, athletes, gaming, footballer, tournaments, athlet, payments, cameras, internet, playing, watches, athletic, camera, cricket, stadium, play, athlete, aired, nfl, golf, advertising, gameplay, storage |
| L23H22 | Mixed | Countries, languages, ethnicity | chinese, china, greek, japanese, japan, beijing, french, russian, italian, spanish, france, mexican, italy, greece, russia, tokyo, greeks, shanghai, russians, finnish, ital, mexico, mex, german, spani, latino, dutch, germany, portuguese, moscow, cyprus, taiwan, brazilian, spain, soviet, ukrainian, germans, swedish, brazil, quebec, guang, hispanic, zhang, jiang, norwegian, ukraine, korean, paris, qing, belgian |
| L26H8 | Mixed | Places, culture, universities | van, dutch, von, las, brazilian, los, filip, french, la, portuguese, han, brazil, hait, france, mexican, lap, italian, mexico, paris, mex, portugal, louis, philippine, spanish, ital, chile, italy, sierra, spain, holland, manila, louisiana, netherlands, so, philippines, jean, monsieur, portug, argentine, barcelona, spani, rio, ucla, argentina, lisbon, haiti, pierre, madrid, brasil, buenos |

Table 11: For a selection of important heads, we display the top 50 tokens that they output (by maximum DLA) from a broader data set with 10,000 prompts. We also include hand written categories that the head specializes in, based on human evaluation of the top 500 tokens that they output. We note that subject, relation and mixed attention heads all seem to specialize to just a few categories.

## F ATTENTION HEAD SUPERPOSITION

Our initial motivation for studying the factual recall set up was to find real world examples of *attention head superposition* (Jermyn et al., 2023). In this section, we explain this motivation.

In mechanistic interpretability, we wish to explain the behavior of neural networks through understanding the representations and algorithms implemented in weights and activations. This requires a notion of the 'fundamental units' of networks. It is a reasonable place to start to investigate the natural structures we find in networks. In some cases, this seems very reasonable: non linear activations produce a privileged basis in the space of neuron activations, which could result in feature representations being aligned to the neuron basis, and individual neurons being interpretable. Unfortunately, we find that neurons are often polysemantic, encoding many different features. We hypothesize this occurs due to superposition: the network is incentivized to encode more features than it has

dimensions. It seems like the correct place to look for features is not in the neurons, but as directions in the neuron activation space. Through a similar argument, we also expect that the residual stream of a transformer stores features in superposition, which is termed *bottleneck* superposition. Much work is being put into the problem of 'solving' superposition, and finding meaningful, interpretable and sparsely activating directions in activation space (Cunningham et al., 2023).

A natural further question to ask is, where else are we studying the wrong fundamental units? In language model interpretability, we often care about localising the computational graph of particular behaviors. This often initially consists of a set of attention heads and MLP layers that "matter" for a given task. But are attention heads themselves the correct unit of study? We know neurons are not, is it possible that attention heads are also not? We have reason to believe the network may try to introduce compression in attention heads themselves. We should expect that models may use a mechanism like this to implement many more behaviors than they have heads. It is possible that each head is individually polysemantic, and implements several distinct behaviors, but in any given context a specific subset of heads work together, attend to the same place, and the output is the residual stream times the weighted sum of their OV matrices. Is there meaningful structure on the set of (n_layers * n_heads) attention heads? Can we productively think of attention heads as being in **superposition** in certain contexts? This idea was first introduced by Jermyn et al. (2023), who suggested attention head superposition as a phenomena, and thought they had found a toy example of it, which they later thought was not quite attention head superposition.

There is some evidence in LLMs for attention head superposition – we often find many heads that seem to be doing the same thing on some sub-distribution. For instance, why are there often several induction heads? In the IOI circuit (Wang et al., 2022), why are there several name mover heads? Can these be productively thought of as a single superposed name mover head? This could additionally explain why negative name mover heads exist. The heads should only be thought of as a single coherent unit, rather than the model learning a real circuit (name movers) and learning a weird anti-circuit (negative name movers) on top.

Here is a theoretical example of head superposition. Say, we have 2 heads X and Y that extract 3 different things (depending on the context) A, B, C. X activations in contexts A and B, giving +A+2B-2C. Y activates for A and C, giving A-2B+2C. Then, in the A context X+Y = 2A, in the B context X+Y = A+2B-2C, and in the C context X+Y = A-2B+2C, and this works. We have compressed 3 tasks into 2 heads. If the "relation propagation" hypothesis of Geva et al. (2023a) were the primary story behind factual recall, factual recall may be a good place to hunt for attention head superposition. Models likely know many more kinds of facts than they have heads, and so may could use heads in combination to extract the correct fact. We however found that models did not use enough relation propagation for this theoretical picture to hold up. Nevertheless, finding examples of attention head superposition is still an interesting future direction.

