# OpenReview forum: "Summing Up the Facts: Additive Mechanisms behind Factual Recall in LLMs"
_ICLR.cc/2024/Conference — Submitted to ICLR 2024_

### Official Review · Reviewer_dGcN · 2023-10-31

**Soundness:** 2 fair
**Presentation:** 3 good
**Contribution:** 2 fair
**Rating:** 5
**Confidence:** 3

**Summary:**

In the context of LLM, this paper shows there exist four distinct and independent mechanisms that additively combine, constructively interfering on the correct attribute. This generic phenomena is termed as the additive motif: models compute correct answers through adding together multiple independent contributions; the contributions from each mechanism may be insufficient alone, but together they constructively interfere on the correct attribute when summed. In addition, this paper extends the method of direct logit attribution to attribute a head’s output to individual source tokens.

**Strengths:**

1. This paper is well written and easy to follow.
2. The experiment is sufficient.

**Weaknesses:**

This finding  seems to be not profound enough. It only demonstrates that LLMs perform better under the additive motif, but it appears insufficient to prove that the additive motif is the underlying factual recall behind LLMs.

**Questions:**

1. This finding may explain the fact that models trained on “A is B” fail to generalize to “B is A”.   Is there any possible to explain the CoT prompting such as “let’s think step by step” by using your findings? Does it bring any other insights or explanations for other phenomena that are difficult to explain in LLMs? For example, is there any possible to explain the CoT prompting such as “let’s think step by step” by using your findings?

2. Can this finding contribute to prompt engineering？

3. Some tables are too wide and are out of page, e.g., table 1.

---

> ### Author Response · Authors · 2023-11-17
>
> Thank you for your review. We’re glad you found the paper to be well written, and were able to follow our experimental results.
>
> > This finding seems to be not profound enough. It only demonstrates that LLMs perform better under the additive motif, but it appears insufficient to prove that the additive motif is the underlying factual recall behind LLMs.
>
> Thanks for raising this – we have provided additional discussion about our main findings, and why we believe them to be novel, in a top level comment to all reviewers. See the sections titled “**additivity**” and “**novelty**”.
>
> > This finding may explain the fact that models trained on “A is B” fail to generalize to “B is A”. Is there any possible to explain the CoT prompting such as “let’s think step by step” by using your findings? Does it bring any other insights or explanations for other phenomena that are difficult to explain in LLMs? For example, is there any possible to explain the CoT prompting such as “let’s think step by step” by using your findings?
>
> Understanding why chain of thought prompting works so well via interpretability is an interesting problem. We however think our set up would not be a good place to study this, due to the simple one step nature of the problem of factual recall, which is fundamentally different to the multi-hop nature of chain of thought prompting.
>
> That said, an interesting area of further investigation may be studying “multi-hop factual recall”, which our results may help explain. Consider prompts of form “The largest church in the world is located in the city of”. Additivity may help models solve this task in one step, even though a human may reason about this problem in a sequential manner. We have added this idea to future work.
>
> > Can this finding contribute to prompt engineering？
>
> We think it’s possible that the insight that models make predictions based on different parts of their input could contribute to prompt engineering. We think this is well known, though has not been shown mechanistically to our knowledge.
>
> For instance, we should expect models to perform worse at factual recall when prompted with “Fact: Michael Jordan plays” over “Fact: Michael Jordan plays the sport of”. We find this to be the case -- Pythia-2.8b reports 21.71% accuracy for the first prompt, yet 69.34% for the second, on the | basketball| token.
>
> In our work, we found that relation heads had meaningful DLA from attending to both the word “plays” and “sport”. Omitting the word “sport” reduces performance. Our work provides a mechanistic explanation for the intuition that providing more detailed information in prompts is beneficial, in at least this context.
>
> We also believe our results could be extended by future work to explain few-shot prompting (e.g. via prepending another related fact in the prompt). Preliminary investigation of this early in the project suggested some of the four mechanisms behaved differently under few shot prompting. For instance, relation heads would fire more strongly on prompts with several mentions of the word “sport”. This is one area of future work we note in the paper.

---

### Official Review · Reviewer_npyu · 2023-11-03

**Soundness:** 3 good
**Presentation:** 3 good
**Contribution:** 2 fair
**Rating:** 5
**Confidence:** 2

**Summary:**

This paper presents a set of experiments on a small hand-crafted data
set for identifying the mechanisms at play during "factual recall" in
large language models. The paper defines four "mechanisms" based on
(1) attention heads that focus (mostly) subject of a factual
predicate, (2) attention heads that focus (mostly) relation, (3)
attention heads that attend to both, and (4) MLP layer. The main claim
is that these mechanisms additively determine the correct attribute.

**Strengths:**

The study tackles an important/interesting problem and the paper reports a substantial amount of experimentation.

**Weaknesses:**

Although I believe the idea is interesting, and there may be some
valuable finding in the paper, I have difficulties seeing a clear
take-home message based on the results presented, and probably also
due to the way they are presented. I have some concrete points of
criticism listed in the comments below (with approximate order of importance).

- The main claim, additivity of the multiple mechanisms, is not very
  clearly demonstrated in the paper. The separation of the
  subject/relation heads (as displayed in Fig. 2) is impressive.
  However, neither the roles of the "mixed head" mechanism, the MLP,
  and additivity of all these mechanisms are not clearly demonstrated.

- The dataset is rather small and it is not described in the paper at
  all. The description of in the appendix is also rather terse,
  containing only a few examples. Given the data set size (hence the
  lack of diversity), and the possible biases (not discussed) during
  the data set creation, it is unclear if the findings can generalize
  or not. In fact, some of the clear results (e.g., the results in
  Fig. 2) may be due to the simple/small/non-diverse examples.

- I also have difficulty for fully understanding the insights the
  present "mechanisms" would provide. To me, it seems we do not get
  any further insights than the obvious expectation that the models
  have to make their decisions based on different parts of the input
  (and meaningful segments may provide independent contributions). I
  may be missing something here, but it is likely that many other
  readers would miss it, too.

- Visualizations are quite useful for observing some of the results.
  However, the discussion of findings based on a more quantitative
  measure (e.g., DLA difference between factual and counterfactual
  attributes) would be much more convincing, precise, repeatable, and
  general.

- Overall, the paper is somewhat difficult to follow, relying data in
  the appendix for some of the main claims and discussion points.
  Appendixes should not really be used for circumvent page-limits.
  Ideally, most readers should not even need to look at them.

- The head type (subject/relation) definition uses an arbitrary
  threshold. Although it sounds like a rather conservative choice, it
  would still be good to know how it was determined.

**Questions:**

Some typo/language issues:
 - Introduction second paragraph: "(Meng et al., 2023a) find ..."
  -> "Meng et al. (2023a) find ..."
- Although it is a very common "mistake" in the field, all
  established style guides I know prescribe that footnote marks
  to be placed after punctuation. Also, I strongly recommend
  against placing footnote marks directly on symbols (like R^3).
- It is a good idea to indicate that figure/table references to
  the appendix are in the appendix.
- The "categories" defined at the beginning of the results section
  comes as a surprise, and seem to be an important part of the
  analysis throughout. This should be defined/explained earlier.
- End of sentence punctuation missing for footnote 4.
- There are no references to Figure 2 from the text.
- It may not be that easy for some figures, but B/W friendly
  figures would be appreciated by people reading on paper or
  monochrome devices (like e-ink readers).
- Some terms like "OV circuit" or "ROME" that many readers are not
  likely to be familiar with should be briefly introduced.
- The same goes for abbreviations of the sort L22H17. Not
  difficult to guess for most readers, but it would be more reader
  friendly to explain at first use.

---

> ### Author Response · Authors · 2023-11-17
>
> Thanks for the generous feedback. We are glad you find our problem choice of understanding how models perform factual recall important, and note that our findings may be valuable to the community.
>
> > The main claim, additivity of the multiple mechanisms, is not very clearly demonstrated in the paper. The separation of the subject/relation heads (as displayed in Fig. 2) is impressive. However, neither the roles of the "mixed head" mechanism, the MLP, and additivity of all these mechanisms are not clearly demonstrated.
>
> We address your concerns regarding the main claim of additivity in a top level comment to all reviewers above titled **additivity**. The clearer distinction for additivity is into two clusters of updates - relating either to the relation or subject. Each of the four model mechanisms contributes to one or both (in the case of mixed heads) of these clusters.
>
>
> > The dataset is rather small and it is not described in the paper at all. The description of in the appendix is also rather terse, containing only a few examples. Given the data set size (hence the lack of diversity), and the possible biases (not discussed) during the data set creation, it is unclear if the findings can generalize or not. In fact, some of the clear results (e.g., the results in Fig. 2) may be due to the simple/small/non-diverse examples.
>
> Thanks for this comment. We have provided further details regarding our dataset in an Appendix C of the paper.
>
> We agree with your criticism regarding the size and potential biases of the dataset. We agree this is not ideal, but found dataset creation difficult. Despite this, our dataset spans several different relations $r$, and contains over 100 prompts, which we think renders our findings valuable and sufficient for demonstration of the *existence* of a range of mechanisms and of additivity. We believe this is a valuable contribution given substantial community interest in both factual recall and interpretability more generally.
>
> We discuss the limitations we faced with dataset creation at some length in Appendix C of the paper.
>
> Quoting from the paper:
>
> ```We found the pre existing datasets to be unsatisfactory for our analysis, due to some additional requirements our set up necessitated. We firstly required models to both \textit{know} facts and to \textit{say facts} when asked in a simple prompting set up, and for the correct attribute $a$ to be completely determined in its tokenized form by the subject and relationship. For example `The Eiffel Tower is in' permits both the answer `Paris' and `France'. For simplicity we avoided prompts of this form. Synonyms also gave us issues, e.g. `football' and `soccer', or `unsafe' and `dangerous'. This mostly restricted us to very categorical facts, like sports, countries, cities, colors etc.  We also wanted to avoid attributes that mostly involved copying, such as `The Syndey Opera House is in the city of Sydney`, as we expect this mechanism to differ substantially from the more general mechanism, and to rely mostly on induction heads \citep{olsson2022context}. Next, we wanted to create large datasets with $r$ held constant, and separately, with $s$ held constant. Holding the relation constant and generating many facts is fairly easy. But generally models know few facts about a given subject, e.g. `Michael Jordan' is associated very strongly with `basketball', but other facts about him are less important and well known. Certain kinds of attributes, like `gender' are likely properties of the tokens themselves, and not likely not reliant on the `subject enrichment' circuitry - e.g. `Michael' and `male'. We try and avoid these cases. We also restrict to attributes where the first attribute token mostly uniquely identifies the first answer token. If the first token of the attribute is a single character or the word “the”, this can be vague, so we omitted these cases. These considerations limited the size of the dataset we studied.```
>
> > I also have difficulty for fully understanding the insights the present "mechanisms" would provide. To me, it seems we do not get any further insights than the obvious expectation that the models have to make their decisions based on different parts of the input (and meaningful segments may provide independent contributions). I may be missing something here, but it is likely that many other readers would miss it, too.
>
> Thanks for this question. We believe this kind of study is valuable in the context of the mechanistic interpretability literature, which we discuss in a top level comment to all reviewers, under the heading “**novelty**”.
>
> *(continued...)*

---

> ### Author Response · Authors · 2023-11-17
>
> > Visualizations are quite useful for observing some of the results. However, the discussion of findings based on a more quantitative measure (e.g., DLA difference between factual and counterfactual attributes) would be much more convincing, precise, repeatable, and general.
>
> Thanks for this comment. We found qualitative figures generally a more useful form to present our results. Our main aim was to demonstrate a range of mechanisms with different qualitative properties exist. This is what is necessary to demonstrate additivity, which is our main goal in the paper.
>
> That said, we do agree that a more raw form of data can also be valuable. In Appendix E we present several instances of such data, on a per-prompt basis. We include for instance tables of raw ordered output tokens, logit ranks, and DLA scores.
>
> > Overall, the paper is somewhat difficult to follow, relying data in the appendix for some of the main claims and discussion points. Appendixes should not really be used for circumvent page-limits. Ideally, most readers should not even need to look at them.
>
> Thank you for the feedback, sorry for this. We have since restructured to emphasise the additivity claim more prominently. This is the main claim in the paper. In order to show additivity, it suffices to have two mechanisms (namely, subject and relation heads). These are distinct, and we explain these in detail. We in fact find four mechanisms, which we include for completeness.  We struggled to fit figures for all four mechanisms in the main body. We therefore made the decision to prioritise “summary” figures in the first results section to give a high level picture of the various mechanisms, and a detailed explanation of two mechanisms.
>
> > The head type (subject/relation) definition uses an arbitrary threshold. Although it sounds like a rather conservative choice, it would still be good to know how it was determined.
>
> Thanks for this comment. We agree this choice is somewhat arbitrary, but we believe it’s useful to draw this distinction. See the top level comment, under “**classifying attention heads**”.
>
> > The "categories" defined at the beginning of the results section comes as a surprise, and seem to be an important part of the analysis throughout. This should be defined/explained earlier.
>
> Thanks for raising this point of confusion. The categories are important to the analysis on the individual component level, but are not conceptually important to the main claim of additivity of the paper. There are several levels of granularity discussed in the paper. From high to low: two additive clusters, four separate groups of mechanisms, and many individual components implementing these. Categories operate on the lowest component level, and merely state individual components (like individual attention heads), have a narrow purpose. We have removed references to categories from the “summary of results” section, to make the main claims regarding additivity flow better and be more clear.
>
> We have additionally addressed all the typos you pointed out - thanks for these.

---

### Official Review · Reviewer_ctMu · 2023-11-07

**Soundness:** 3 good
**Presentation:** 3 good
**Contribution:** 2 fair
**Rating:** 5
**Confidence:** 4

**Summary:**

This work target at interpreting the inner mechanisms of LLMs in accomplishing the task of Factual Recall. This work identifies and explains four distinct mechanisms present in the model, as well as the additive cooperation between these mechanisms. This work validates the generalizability of this mechanism across different models and facts.

**Strengths:**

(1) Based on sufficient experimental results verification, the author has identified and explained the internal mechanisms of LLMs at the granularity level of attention heads and MLPs. More interestingly, it provides an explanation of the “reversal curse” phenomenon discovered in recent works.
(2) This work has thoroughly discussed the related work and proposed a range of possible directions for future works.

**Weaknesses:**

(1) There have been many works [1, 2] interpreting the model behavior of Factual Recall. It seems that the novelty is insufficient with only a deeper zooming into attention heads using similar interpretability methods. Additionally, the discovery of the additive motif is not surprising enough, as already explained in work [3] that "Attention heads can be understood as independent operations, each outputting a result which is added into the residual stream."
(2) Is direct logit attribution (DLA) the same as the interpretability method of Path Patching [4] or Causal Mediation Analysis [5]? If so, it is necessary to explain how the counterfactual data is applied for causal intervention. If it is not, it is necessary to provide a detailed description of the algorithm flow of DLA.

(3) This work extends “DLA by source token group” with a weighted sum of outputs corresponding to distinct attention source position. But how to obtain the “weights”? How to attribute multiple tokens simultaneously? These missing implementation details make it difficult to understand the method and reproduce the results.


[1] Locating and Editing Factual Associations in GPT
[2] Dissecting Recall of Factual Associations in Auto-Regressive Language Models
[3] A Mathematical Framework for Transformer Circuits
[4] Interpretability in the Wild: a Circuit for Indirect Object Identification in GPT-2 small
[5] Investigating Gender Bias in Language Models Using Causal Mediation Analysis

**Questions:**

(1) It would be better to validate the faithfulness of the identified components (e.g., Subject Heads, Relation Heads) for Factual Recall? What would happen to the prediction ability (e.g., accuracy) of the model for Factual Recall task if these components were knocked out?

(2) We wonder if it is possible to explain the behavior of MLPs explicitly, similar to explaining Attention Heads via attention patterns?

---

> ### Author Response · Authors · 2023-11-17
>
> Thanks for the detailed review. We’re glad that you found our experimental results sufficient to explain the internal mechanisms LLMs use to recall facts. We’re also happy to hear that you find the work relevant in the context of the existing literature. Below we comment on some of the weaknesses you point out in the paper.
>
> > (1) There have been many works [1, 2] interpreting the model behaviour of Factual Recall. It seems that the novelty is insufficient with only a deeper zooming into attention heads using similar interpretability methods.
>
> Our approach is fundamentally different to [1] and [2], and our findings subsequently different. As you noted, we chose to zoom in deeply into individual model components, which [1] did not do at all, and [2] only briefly studied. Through using a “circuits style” approach, we were able to find insights that both [1] and [2] missed. [1] studies where factual knowledge is stored in models, but not how such knowledge is used to predict the next token. [2] attempts to study this problem, but only does so through coarse grained ablations. They suggest a mechanism based on this, which we search for in individual model components. In doing so, we instead find several independent mechanisms that explain how information regarding factual knowledge is moved. We additionally find an extra place where factual information is stored – namely in the relationship token “sport” itself - prior work missed this important aspect, which we found through careful study of a range of counterfactual attributes S and R. This motivated the focus on “additivity” (see the top level comment titled "**additivity**" for more discussion on this). We also discuss the "**novelty**" of this work in another top level comment.
>
> > Additionally, the discovery of the additive motif is not surprising enough, as already explained in work [3] that "Attention heads can be understood as independent operations, each outputting a result which is added into the residual stream."
>
> Thanks for raising this confusion. We address what we mean by additivity in a top level comment to all reviewers, and have amended the paper to clarify this. Our use of additivity is distinct from [3].
>
> > (2) Is direct logit attribution (DLA) the same as the interpretability method of Path Patching [4] or Causal Mediation Analysis [5]? If so, it is necessary to explain how the counterfactual data is applied for causal intervention. If it is not, it is necessary to provide a detailed description of the algorithm flow of DLA.
>
> Thanks for raising this confusion. Direct logit attribution (DLA) is not the same technique as path patching or causal mediation analysis. DLA is a simple technique that we describe in the paper in section 2. We have now added further discussion on this, including a detailed description of the algorithm, in Appendix D. It is not based on a causal intervention. It builds on a technique named the “logit lens”. Both are based on the insight by [3] that the residual stream is an accumulated sum of model components, and that the map to logits from the final residual stream vector is approximately linear. While running a single forward pass, we may take the output from individual model components, such as attention heads, and directly map these to logit space, by immediately applying the model unembedding (aka language model decoder head/final linear layer). Given a set of logits, we can read off the DLA for any possible output token, including for counterfactual output tokens.
>
> > (3) This work extends “DLA by source token group” with a weighted sum of outputs corresponding to distinct attention source position. But how to obtain the “weights”? How to attribute multiple tokens simultaneously? These missing implementation details make it difficult to understand the method and reproduce the results.
>
> Our extension is to consider the attention head output as a weighted sum over individual source positions, where the weighting is given by the attention probability. This is entirely faithful to transformer computation. By unravelling this sum, we can consider DLA for attention heads to be attributed to all attention source tokens individually. We have provided a detailed discussion with formulae of this technique in Appendix D.
>
> > (1) It would be better to validate the faithfulness of the identified components (e.g., Subject Heads, Relation Heads) for Factual Recall? What would happen to the prediction ability (e.g., accuracy) of the model for Factual Recall task if these components were knocked out?
>
> Thanks for this question. We address this question in the top level comment, under “**ablations**”.
>
> *(continued...)*

---

> > ### Author Response · Authors · 2023-11-17
> >
> > > (2) We wonder if it is possible to explain the behaviour of MLPs explicitly, similar to explaining Attention Heads via attention patterns?
> >
> > Explaining the behaviour of MLPs is well documented to be significantly harder than interpreting attention heads in the interpretability literature. The function of attention heads can be thought of as primarily of moving information around different token positions. MLPs instead perform meaningful computation, and have many more parameters. They have also been noted to operate on features under a large amount of superposition [1] – meaning individual neurons encode many different concepts and are in general uninterpretable in networks in isolation. In this work, we only attempt to interpret the entire output of MLP layers, but agree that further work could investigate this computation more deeply. As this is not central to the key claims of the paper, we note this as a possible area of future investigation in the future work section.
> >
> > [1] FINDING NEURONS IN A HAYSTACK: CASE STUDIES WITH SPARSE PROBING

---

> > > ### Comment · Reviewer_ctMu · 2023-11-22
> > > **Re: authors response**
> > >
> > > After carefully read the reply and revised appendix in the paper, I think my concern about the novelty and the takeaway contributions of this paper is still not clear enough, I decide to lower my score from 6 to 5.

---

> > > > ### Author Response · Authors · 2023-11-22
> > > >
> > > > We're surprised and saddened that you've decided to lower your score. Can you elaborate on what led you to change your mind from your initial review, so we can better engage with the critique?
> > > >
> > > > We believe we've actually improved the paper in terms of novelty, by making clearer what the additive motif means, why it's important, and why prior work missed this. We have made changes in the paper at various points to better reflect this (e.g. in the Introduction, Results, Conclusion), and have remarked on it at length in the top level comment.

---

### Official Review · Reviewer_vZgL · 2023-11-09

**Soundness:** 2 fair
**Presentation:** 3 good
**Contribution:** 2 fair
**Rating:** 6
**Confidence:** 2

**Summary:**

The work concerns the task of factual recall in LLMs i.e. in templated prompts, the LLM is tasked to predict the object attribute of the tuple (subject, relation, attribute). Authors propose that factual recall in the END position (correct logit ranking) occurs by the summation of contributions of different additive circuits in the transformer.
- Authors extend Direct Logit Attribution (DLA) to compute the joint contribution from different source token groups to the final predicted logits
- 4 different additive circuits are identified based on the extended DLA: SUBJECT, RELATION, MIXED and MLP
- SUBJECT attention heads preferentially boost attributes that are relevant to the subject of the query
- RELATION attention heads preferentially boost attributes that are relevant to the relation of the query independent of the subject
- MIXED attention heads boost the attributes that are jointly relevant to the subject and relation of the query
- MLP layers at the end position uniformly boost the attributes relevant to the relation (ignoring the subject tokens)

The central findings of the paper revolve around the Pythia-2.8b model. Additional experiments in the Appendix report that similar types of circuits may be found in other models but all categories may not always exist.

Limitations:
- Authors acknowledge that the boundary between MIXED and other attention head types is fuzzy. The definition used to separate attention heads was based on preferential contribution from SUBJECT or RELATION and any other type is considered a mixed type.

**Strengths:**

- The paper uses established mechanistic interpretation tools and extends them to identify mechanisms in the transformer that perform very specific purposes
    - The SUBJECT-head, RELATION-head, and MLP additive behaviors are established by showing consistent patterns across a range of fact queries

**Weaknesses:**

- The paper introduction and further discussions claim that the results reported here provide a mechanistic explanation for the limitations of LLMs to learn "B is A" from training on "A is B" [1]. However, I do not see sufficient evidence to support this claim
    - They have shown that in the forward direction the transformer selectively promotes attributes relevant to the subject and the relation
    - This does not show that the transformer CANNOT/DOES NOT perform the same operations in the reverse direction.
    - E.g. "Basketball is played by ..." may contain circuits that selectively promote the known basketball players. The lack of such circuits is not demonstrated by this work
    - In particular, the authors argue that the LLM learns an "asymmetric" look-up. However, the asymmetry is not established.

Presentation
---
- Significant space in the main paper is used to describe future work. I believe that there is an interesting and valuable discussion about dataset creation in the Appendix that should be brought to the main paper


[1] Lukas Berglund, Meg Tong, Max Kaufmann, Mikita Balesni, Asa Cooper Stickland, Tomasz Korbak, and Owain Evans. The Reversal Curse: LLMs trained on "A is B" fail to learn "B is A", September 2023. URL http://arxiv.org/abs/2309.12288. arXiv:2309.12288 [cs].

**Questions:**

1. Is it fair to say that the key findings are the presence of SUBJECT-only and RELATION-only heads among the attention heads in the transformer? All other heads are MIXED heads by default?
2. What fraction of attention heads get categorized into extreme categories (SUBJECT and RELATION)?
3. How does the contribution to the final logits from the extreme categories (SUBJECT and RELATION) compare to the heads that are categorized as MIXED?
4. Tagging onto questions 3 and 4: is there a significant drop in model performance when extreme heads are suppressed?

---

> ### Author Response · Authors · 2023-11-17
>
> Thanks for the generous review. We are glad you appreciated our use of existing and novel mechanistic interpretability techniques to identify mechanisms with specific purposes used for factual recall in transformer language models.
>
> > The paper introduction and further discussions claim that the results reported here provide a mechanistic explanation for the limitations of LLMs to learn "B is A" from training on "A is B" [1]. However, I do not see sufficient evidence to support this claim
>
> We thank the reviewer for their feedback. We believe that our work does provide a mechanistic explanation for the reversal curse, but thank the reviewer for pointing out the need to communicate this more clearly. The reversal curse claims that models trained on “A is B” fail to generalise to “B is A”. Importantly, this is difficult to verify with purely pre-trained models on common facts (e.g. Michael Jordan plays basketball) where the model likely saw the fact in both forms (basketball is played by Michael Jordan, Michael Jordan plays basketball). The reversal curse paper demonstrates this with fine-tuning on novel facts, ensuring the reverse-direction was not seen.
>
> In our work, we only study a pre-trained model, and so the evidence we provide is indirect and suggestive, as we do not fine-tune ourselves. We find a circuit by which models may learn to output “A is B”, involving subject enrichment on the A tokens, and some attention head attending to A and extracting B. Importantly, this is a unidirectional circuit with two unidirectional components - it extracts the fact “B” from “A”.  This suggests that the reason fine-tuning on “A is B” does not boost “B is A” in general is because training on “A is B” only boosts the unidirectional A -> B mechanisms, and has no effect on potential B -> A mechanisms.
>
>
> > Is it fair to say that the key findings are the presence of SUBJECT-only and RELATION-only heads among the attention heads in the transformer? All other heads are MIXED heads by default?
>
> Yes, in part. We believe our key finding is that there exist a range of different mechanisms with qualitatively different functions that all contribute to this task, and which interact additively. We clarify what we mean by this in the top level comment. All other heads are mixed, yes. Some heads are of course more important than others for this subtask.
>
>
> > What fraction of attention heads get categorized into extreme categories (SUBJECT and RELATION)?
>
> > How does the contribution to the final logits from the extreme categories (SUBJECT and RELATION) compare to the heads that are categorized as MIXED?
>
> > Tagging onto questions 3 and 4: is there a significant drop in model performance when extreme heads are suppressed?
>
>
> Thanks for these questions regarding the prevalence and importance of the various mechanisms. All three of these questions ask similar questions. We address the first two of your questions here, and refer you to the top level comment “ablations” for an answer to the third.
>
> The **fraction of heads** classified each way varies depending on the choice of relation. See Figure 3 for two examples. There, we see the split of (subject, relation, mixed) among the top 10 heads for the relation “plays the sport of” is (2, 1, 7), but for “is in the country of” it is (4, 2, 4). In response to this question, we ran experiments to check what the split was across our entire dataset and without aggregation over the relation. Inspecting the top 10 heads by DLA for each example we find 37% of heads get categorised as subject heads and 33% as relation heads, with the remaining 30% as MIXED heads. This indicates all three head types are important for the task.
>
> The **contribution to logits** is another good metric. Figure 2 visualises this – we can qualitatively see that all three head types are important. In response to this question, we ran some additional experiments. We include below the percentage of the final (mean centred) logit contributed from each component type, across the entire dataset. We omitted several negative suppressive components for the purpose of this analysis. Again, we see that the contributions from each type of mechanism is important. Subject heads contribute 18%, relation heads 24%, mixed heads 27%, and the mlp layers 30%, across the entire dataset.
>
> *(continued...)*

---

> > ### Author Response · Authors · 2023-11-17
> >
> > > Authors acknowledge that the boundary between MIXED and other attention head types is fuzzy.
> >
> > Thanks for this comment. We provide some discussion in the top level comment under the heading “**classifying mechanisms**”.
> >
> > > Significant space in the main paper is used to describe future work. I believe that there is an interesting and valuable discussion about dataset creation in the Appendix that should be brought to the main paper
> >
> > Thanks for this comment on the limitations section. We agree these limitations are very helpful. We have since added significant further discussion to the main body, so no longer have space to include this section in the main body. We have pointed to it more visibly from the main text in the methods section.

---

### Author Response · Authors · 2023-11-17

We extend our thanks to all reviewers for their insightful reviews and valuable questions. We have responded to each reviewer individually, and made corresponding changes in the paper in our revised version of the paper in **red**.  We welcome further responses and questions from the reviewers. We are keen to answer any remaining unanswered questions and to hear of further ways to improve the work. In the remainder of this top level comment, we address some common points among reviewer questions.

## Additivity

In this paper, our core contribution is showing that factual recall is additive. Reviewers **npyu** and **dGcN** felt this was unclear in our manuscript. In this part of the response, we address these concerns. We will first give a definition of additivity. We then discuss some examples of additivity in both toy set ups, and in factual recall. We finally discuss how our empirical results demonstrate additivity of factual recall in the paper. We have made modifications to the paper’s introduction and results sections to make our claims regarding what additivity is, and how we show it, more clear. We have also reordered parts of the paper to highlight our key contributions more prominently.

We say models produce outputs additively if
1) There are multiple important model components whose outputs independently directly boost the correct logit.
2) These components are qualitatively different – their distribution over output logits are meaningfully different.
3) These components constructively interfere on the correct answer, even if the correct answer is not the argmax output logit of components in isolation.
While it may seem noisy and unreliable to stack heuristics like this, softmax extremises outputs due to the exponential, which makes this strategy less disadvantageous to loss.

As a toy example, consider a logistic model that is tasked with predicting whether an integer is divisible by 6 (into two classes, true or false). Consider the following two mechanistic ways of solving the task
a) Solve the task directly, memorising which integers are divisible by 6.
b) Solve the task in two independent parts. Assign a +1 true logit to all numbers divisible by 2. Assign +1 true logit to all numbers divisible by 3, with a different circuit. Apply a uniform bias producing a  -1.5 false logit.
(a) is non additive. (b) is additive, by the criteria (1-3). There are two different components that contribute to the answer (1), they have qualitatively different outputs (2), and they constructively interfere on the correct answer, with each component insufficient alone (3). Note that condition (2) is necessary to exclude cases where the model increases its confidence through adding two components with identical outputs, which we do not consider to be additive. This example is analogous to how a transformer functions, since the residual stream is an additive sum of outputs from model components, and there’s an (approximately) linear map from the residual stream to the output logits given by the unembedding (aka the language modelling decoder head) [1], so each component can be considered to be writing to logits separately in a linear fashion.

Now, we move on to the set up in the paper. Consider the example shown in Figure 1, of completing “The Colosseum is in the country of”. There are two sources of information here - the subject “Colosseum” and the relation “country”. These correspond to two clusters of additive updates - updates that write many attributes about the Colosseum (“Italy”, “Rome”, “amphitheatre”, …) and updates that write many countries (“Italy”, “USA”, …).

Our main claim in the paper is that factual recall is performed additively in this way. We set out to show this by studying model internals. Our empirical results support additivity. We find four mechanisms that are all implemented differently by the model. The outputs of these mechanisms group into the two clusters we describe above. Each mechanism independently boosts the correct answer (condition 1). There are two qualitatively different clusters of output behaviour (condition 2). This is mostly clearly shown in Figures 2 and 3. Subject and relation heads have very different output behaviour (via Direct Logit Attribution (DLA) [1, 2, 3], on both correct and counterfactual attributes).

(continued...)

---

> ### Author Response · Authors · 2023-11-17
>
> **ctMu** asks how this relates to the finding of [1], that provides a mathematical framework for thinking about transformer computation. In this work, the authors argue that the key object of interest in a transformer is the so-called “residual stream”, from which every model component reads and writes. This is an alternate but mathematically identical way of reasoning about skip connections. The individual model components that read and write to this residual stream are neurons in MLP layers and attention heads, all of which are independent. These contributions are therefore additive, in some sense. This is not quite what we mean by additive, as described above, but is related.
>
> ## Novelty
>
> Reviewers **npyu** and **dGcN** raised concerns that our key finding is not profound or novel enough. While it is true that we should expect models to use multiple parts of their input for prediction, our work makes this claim rigorous on a mechanistic level, which prior work had not attempted to do. Our work also highlights a particular way in which this information is combined. The model could use the two sources of information together, as part of a larger compositional circuit (e.g. use the relation information to select an attribute from the subject to extract). We instead find the additive motif as described above is primarily used, which has so far not been documented. In particular, something of the form of “relation heads”, can be thought of as setting the correct reference class for answers (e.g. “sports”). The study of this is neglected in mechanistic circuit analysis in general. We have improved our exposition of this contribution in the paper.
>
> For instance, in the work by Wang et al. [6], models are tasked with completing sentences of the form “When John and Mary went to the store, John bought flowers for”. This task has two components –  (a) figure out the answer should be a name, and then (b) figure out what the correct name is. In [2], the authors use the metric of “logit difference”, “Mary - John”, and isolate the circuit for (b), but neglect to study (a). While just studying (b) is valid, it is important to be explicit that part of the behaviour remains unexplained. Our work instead claims that (a) is an important part of predicting the next token, especially in factual recall, and so we study it on a mechanistic level. This is novel among the mechanistic interpretability literature.
>
> Such considerations are also neglected within the factual recall literature. Prior work did not study many other individual counterfactual outputs as we did with sets S and R, so missed the fact that many other attributes were positively upweighted by the model. Our work shows single facts are not extracted, but many different facts are extracted. Figuring out how the model decides between these is an interesting question, which we explore. This makes our findings novel among the existing work on how transformers recall facts.
>
> (continued...)

---

> ### Author Response · Authors · 2023-11-17
>
> ## Ablations
>
> Reviewers **vZgL** and **ctMu** were curious what the effect on performance would be upon knocking out various mechanisms. We originally chose not to perform ablation experiments, as we had already studied the direct effect of model components through their Direct Logit Attribution [DLA], and reasoned ablation experiments would not be much more informative.
>
> In response to this feedback, we decided to run some ablation experiments. We have included these results in Appendix E.3. Naive ablations have been noted in prior work to be counteracted by self-repair in the factual recall set up, a phenomenon known as the Hydra Effect [4]. We therefore followed the approach of [6], and performed edge patching - ablating the direct path term between model components and logits. We present baseline loss, together with the loss after knocking out one model mechanism. Each loss reported is aggregated over a dataset with fixed relation $r$. We see knocking out any individual mechanism significantly harms loss in each case.
>
> | Relation       | Baseline Loss | Subject % Change | Relation % Change | Mixed % Change Loss | MLP % Change |
> |----------------|---------------|------------------|-------------------|---------------------|--------------|
> | PLAYS_SPORT    | 0.68          | 16.71            | 254.31            | 345.56              | 483.11       |
> | IN_COUNTRY     | 0.51          | 250.29           | 710.23            | 375.23              | 207.43       |
> | CAPITAL_CITY   | 1.41          | 67.38            | 206.48            | 90.53               | 222.64       |
> | LEAGUE_CALLED  | 1.58          | 170.37           | 97.31             | 185.97              | 104.80       |
> | PROFESSOR_AT   | 0.62          | 127.90           | 503.27            | 78.52               | 714.40       |
> | PRIMARY_MACRO  | 1.60          | 190.55           | 10.92             | 69.08               | 189.48       |
> | PRODUCT_BY     | 0.75          | 112.21           | 578.38            | 145.64              | 249.04       |
> | FROM_COUNTRY   | 1.16          | 196.88           | 234.61            | 101.40              | 255.63       |
>
> (continued...)

---

> > ### Author Response · Authors · 2023-11-17
> >
> > ## Classifying attention heads.
> >
> > Reviewers **vZgL** and **npyu** wanted to hear about how we chose our classification of attention heads. We chose a fairly conservative figure arbitrarily and then verified it matched out intuitions for what such heads do by inspecting many versions of Figure 3. Among the most important few heads, we found our definition to make sense. Our definition was in terms of DLA, but we also verified this lined up with where attention was paid.
> >
> > We agree that the fuzziness of the boundary behind particular mechanisms is a limitation of our framing of attention heads as fitting into one of three distinct categories. In reality, there is some sliding scale between a head acting as a relation head and subject head. We chose this classification as we found it helpful in understanding model behaviour. We view one of our contributions as showing there *exist* heads with different qualitative behaviour in the end to end factual recall circuit, which we believe this framing illuminates well.
> >
> >
> > [1] A Mathematical Framework for Transformer Circuits.
> >
> > [2] interpreting GPT: the logit lens
> >
> > [3] Analyzing Transformers in Embedding Space
> >
> > [4] The Hydra Effect: Emergent Self-repair in Language Model Computations
> >
> > [5] Causal Scrubbing: a method for rigorously testing interpretability hypotheses [Redwood Research]
> >
> > [6] Interpretability in the Wild: a Circuit for Indirect Object Identification in GPT-2 Small

---

### Author Response · Authors · 2023-11-21

With the discussion period nearing its close, we would like to remind the reviewers that have only left initial reviews that we would appreciate hearing further feedback on our response. We are keen to know if our responses have adequately addressed all their concerns. Should there be any outstanding queries, we are also eager to engage further.

---

### Comment · Reviewer_npyu · 2023-11-23

Although I was not able to participate in the extensive discussion timely. I did read all the comments and reread the changes to the paper. The paper, after the revisions, definitely improved particularly for clarity. I still have doubts about the exact contribution of the study. I have, nevertheless, edited my review to reflect the improvements on the clarity.

---

### Meta-Review · Area_Chair_ZTLs · 2023-12-10

**Metareview:**

This paper presents a series of experiments conducted on a small, manually curated data set designed to identify the mechanisms for "factual recall" in large language models. The paper defines four distinct "mechanisms," which are based on (1) attention heads that primarily focus on the subject of a factual predicate, (2) attention heads that primarily focus on the relation between the subject and object, (3) attention heads that attend to both the subject and relation, and (4) multi-layer perceptron (MLP) layers. The central claim of the paper is that these mechanisms work together additively to determine the correct attribute. The problem studied in the paper and the findings presented are of interest to the community. As several reviewers pointed out, the presentation of the paper should be significantly improved before the paper is ready for publication. In addition, the datasets used in the analysis should be strengthened to make the results more convincing.

**Justification For Why Not Higher Score:**

As several reviewers pointed out, the presentation of the paper should be significantly improved before the paper is ready for publication. In addition, the datasets used in the analysis should be strengthened to make the results more convincing.

**Justification For Why Not Lower Score:**

N/A

---

### Decision · Program_Chairs · 2024-01-16

Reject